# LATENT TRAJECTORY: A NEW FRAMEWORK FOR ACTOR-CRITIC REINFORCEMENT LEARNING WITH UNCERTAINTY QUANTIFICATION

## ABSTRACT

Uncertainty quantification for deep neural networks is crucial for building reliable modern AI models. This challenge is particularly pronounced in deep reinforcement learning, where agents continuously learn from their interactions with stochastic environments, and the uncertainty of the value function is a key concern for ensuring reliable and robust RL applications. The complexity increases in actor-critic methods, as the training process alternates between optimizing the actor and critic networks, whose optimization nature makes the uncertainty of the value function hard to be quantified. To address this issue, we introduce a novel approach to RL training that conceptualizes transition trajectories as latent variables. Building on this framework, we propose an adaptive Stochastic Gradient Markov Chain Monte Carlo (SGMCMC) algorithm for training deep actor-critic models. This new training method allows for the implicit integration of latent transition trajectories, resulting in a trajectory-independent training process. We provide theoretical guarantees for the convergence of our algorithm and offer empirical evidence showing improvements in both performance and robustness of the deep actor-critic model under our Latent Trajectory Framework (LTF). Furthermore, this framework enables accurate uncertainty quantification for the value function of the RL system, paving the way for more reliable and robust RL applications.

## 1 INTRODUCTION

Reinforcement learning (RL) solves sequential decision-making problems by designing an agent that interacts with its environment to learn an optimal policy, with the goal of maximizing the value function, i.e., total amount of expected rewards. Therefore, accurately quantifying the uncertainty of the value function has been a critical concern for ensuring reliable and robust RL applications. However, achieving this goal in the context of deep RL is notably challenging, despite the universal approximation capabilities of deep neural networks (DNNs) having significantly expanded the applicability of RL. The primary difficulty in uncertainty quantification for deep RL arises from two aspects: the complex architecture of DNNs and the adaptive nature of the RL process. Even within the context of supervised learning, accurately quantifying uncertainty for DNNs is a challenging problem (see, e.g., (Blundell et al., 2015) and (Lakshminarayanan et al., 2017)), and the adaptive process in RL further complicates this issue (Osband et al., 2016; Bellemare et al., 2017). A significant step toward addressing this challenge has been made in Shih & Liang (2024), where deep $Q$-networks are simulated from their posterior distribution under the Kalman temporal difference (KTD) framework (Geist & Pietquin, 2010; Shashua & Mannor, 2020), enabling accurate quantification for the uncertainty of the Q-value function throughout the RL process. However, their methods are challenging to extend to deep RL settings involving actor-critic architectures. The inclusion of an additional actor network significantly complicates the process of uncertainty quantification, as the actor's stochastic policy directly influences the distribution of the critic's value estimates, creating interdependencies that are not easily captured by traditional approaches.

To address the challenge of uncertainty quantification in actor-critic models, we propose a novel framework for deep RL training, which conceptualizes transition trajectories in the RL process as latent variables. Building on this perspective, we introduce an adaptive Stochastic Gradient Markov

Chain Monte Carlo (SGMCMC) algorithm, which simultaneously updates the actor network through a stochastic gradient descent (SGD) step and samples from the conditional distribution of the critic network — conditioned on the current actor network — via an SGMCMC step. Under mild regularity conditions, we establish the convergence of this adaptive SGMCMC algorithm. Specifically, we show that the parameters of the actor network converge in probability to a fixed point, while the parameters of the critic network converge weakly to a target distribution, thereby enabling accurate quantification of the associated value function's uncertainty. In summary, our study offers two primary contributions:

- We introduce a novel latent trajectory framework for training deep actor-critic models that inherently accounts for the dynamic nature of complex RL processes.

- We establish a convergence theory for the SGMCMC algorithm used for training deep actor-critic models under the latent trajectory framework, which ensures effective training for the actor network while enabling proper uncertainty quantification for the critic network and thus the value function.

To our knowledge, this is the first work for training deep actor-critic models with uncertainty quantification.

**Related Works**   Bayesian methods has been generally considered as a standard approach for uncertainty quantification in machine learning. However, a rigorous implementation of Bayesian methods for online RL is challenging. Specifically, Bayesian methods aim to infer unknown model parameters (denoted by $\theta$) based on their posterior distribution $\pi(\theta|D)$ for a given dataset $D$ of identically distributed samples; however, in online RL, the data are drawn from a dynamic system where the samples are non-identically distributed. Furthermore, under the deep learning setting, it is challenging to specify the prior distribution that ensure the posterior consistency property holds when the deep neural networks involve more parameters than the training sample size. While there is existing work on Bayesian RL, such as Fellows et al. (2024) and Osband et al. (2018), these challenges are not adequately addressed. For instance, Osband et al. (2018) introduces uncertainty through randomized priors in a maximum a posteriori (MAP) framework for approximate inference. However, the theoretical property of posterior consistency for such randomized priors has not been established. The actor-critic architecture introduces further complexity due to the interaction between the actor and critic networks, which exacerbates the challenges in uncertainty quantification.   Some other methods, such as bootstrapping (Osband et al., 2016; Tasdighi et al., 2024), Gaussian processes (Geist & Pietquin, 2010; Engel et al., 2003), distribution RL (Bellemare et al., 2017), have also been proposed for uncertainty quantification in online RL. However, their theoretical guarantees remain unestablished in the context of the deep actor-critic setting. In this paper, we propose an innovative and theoretically rigorous framework for uncertainty quantification in online RL. Our approach explicitly models the dynamics of online RL, ensuring that the uncertainty in the critic network is accurately captured, even under the challenges posed by non-identically distributed data and multiple deep neural networks.

## 2   PRELIMINARIES ON ACTOR-CRITIC MODELS

We consider discounted, finite horizon policy optimization problems. Let $\theta$ and $\psi$ denote the parameters of the actor and critic networks, respectively. Let $(s_0, a_0, s_1, a_1, \dots)$ be the transition trajectory generated by a stochastic policy $\pi_\theta$, where each action $a_t$ is sampled from the distribution $\pi_\theta(a_t|s_t)$. At each time step $t$, the agent receives an immediate reward $r_t = r(s_t, a_t)$. Let $R_t = \sum_{\tau=t}^{T-1} \gamma^{\tau-t} r_\tau$ be an unbiased estimate of the Q-value, denoted by $Q^{\pi_\theta}(s_t, a_t)$. Let $V_\psi$ be the critic network approximation to the value function $V^{\pi_\theta}$. For convenience, we denote a single transition of the state and action as $x = (s, a)$, the return estimate as $R$. In this paper, we focus on the advantage actor-critic algorithm (Sutton et al., 2000; Schulman et al., 2018) with the advantage function expressed as:

$$A_\psi(s_t, a_t) = R_t - V_\psi(s_t), \tag{1}$$

where $A_\psi$ indicates the dependence of the advantage function on the critic network $\psi$. The policy gradient (Sutton & Barto, 2018) for the advantage actor-critic algorithm is then given by:

$$g_\psi^{ac}(\theta) = \mathbb{E}_{\pi_\theta}[\sum_{t=0}^{T-1} A_\psi(s_t, a_t)\nabla_\theta \log \pi_\theta(a_t|s_t)]. \tag{2}$$

Note that different parameterization strategies can be employed for the advantage function. For example, one can parameterize the V-function, and use temporal difference (TD) or Monte Carlo methods to estimate the Q-function (Schulman et al., 2018). The parameters $\theta$ and $\psi$ are then iteratively updated using stochastic gradient optimization algorithms till convergence. However, the convergence theory for such an iterative optimization algorithm is hard to be established except for special cases under restrictive assumptions, such as linear function approximation (Chen et al., 2023; Wu et al., 2022), greedy policies (Holzleitner et al., 2020). In practice, many implementations have been proposed, such as A2C, A3C(Mnih et al., 2016), PPO(Schulman et al., 2017), SAC(Haarnoja et al., 2018), and DDPG(Lillicrap et al., 2019), which employ different tuning techniques for policy gradients to enhance the convergence and the stability.

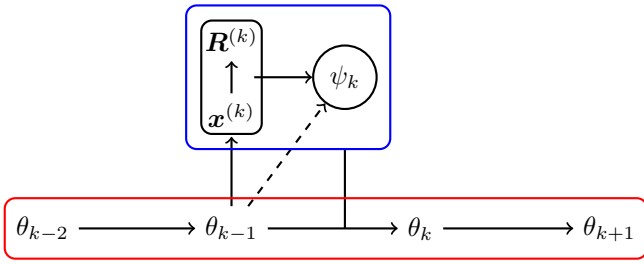

Figure 1: Latent Markov Sampling process, where $\psi_k$ is conditionally independent of $\theta_{k-1}$ given $\boldsymbol{x}^{(k)}$, and the dashed line indicates that including the latent variables $(\boldsymbol{x}^{(k)}, \boldsymbol{R}^{(k)})$ breaks the original dependence between $\psi_k$ and $\theta_{k-1}$.

Consider the actor-critic training process illustrated in Figure 1. Let $k$ index the updates of $\theta$ during training, and let $\boldsymbol{x}^{(k)} = \{x_i^{(k)}\}_{i=1}^n$ denote a batch of transition tuples drawn independently from the stationary distribution $\pi(x|\theta_{k-1})$, where $x_i^{(k)} = (s_i^{(k)}, a_i^{(k)})$ and $n$ is the batch size. Additionally, let $\boldsymbol{R}^{(k)} = \{R_i^{(k)}\}_{i=1}^n$ be the estimated returns corresponding to $\boldsymbol{x}^{(k)}$. At each iteration $k$, a transition trajectory of size $n$ is generated from the policy $\pi_{\theta_{k-1}}$, $\psi_k$ is then updated based on the trajectory. Due to the stochastic nature of the RL process, $\psi_k$ can be viewed as a sample drawn from the conditional distribution $\pi(\psi_k|\theta_{k-1})$. For mathematical clarity, we assume that $\boldsymbol{x}^{(k)}$ is sampled from a pseudo-population of size $\mathcal{N}$, while the pseduo-pupulations can vary for different values of $\theta_{k-1}$. In what follows, we use $\pi_{\mathcal{N}}(\psi|\theta)$ to denote the conditional distribution of $\psi$ for a given value of $\theta$. The concept of pseudo-population allows for the flexibility of using different mini-batch size at different iterations, which determines the accuracy we can reach in inferring the target conditional distribution $\pi(\psi|\theta)$.

The ultimate goal of RL is to learn an optimal policy, with transition data serving as an intermediate step in this process. Thus, the transition trajectory $(\boldsymbol{x}^{(k)}, \boldsymbol{R}^{(k)})$ and the value function parameter $\psi_k$ can be naturally treated as latent variables that facilitate policy optimization.This perspective allows us to frame the training of the actor network in terms of solving the following integral equation:

$$g(\theta) = \int g_\psi^{ac}(\theta)\pi_{\mathcal{N}}(\psi|\theta)d\psi = 0, \tag{3}$$

which can be solved using an adaptive SGMCMC algorithm (Liang et al., 2022a; Deng et al., 2019). This will be detailed in Section 3.

## 3 A LATENT TRAJECTORY FRAMEWORK FOR ACTOR-CRITIC MODELS

### 3.1 AN OVERVIEW OF THE SGMCMC ALGORITHM

To solve the equation (3), an adaptive SGMCMC algorithm consists of two steps at each iteration:

1. ($\psi$-sampling) Simulate $\psi_k \sim \pi_{\mathcal{N}}(\psi|\theta_{k-1})$ by a SGMCMC algorithm.

2. ($\theta$-updating) Update $\theta_k = \theta_{k-1} + \omega_k \hat{g}^{ac}_{\psi_k}(\theta_{k-1})$, where $\omega_k$ denotes the step size used in the stochastic approximation procedure (Robbins & Monro, 1951), and $\hat{g}^{ac}_{\psi_k}(\theta_{k-1})$ is an unbiased estimate of $g^{ac}_{\psi_k}(\theta_{k-1})$.

Under mild conditions, we establish the convergence of the proposed algorithm. Specifically, we show that $\|\theta_k - \theta^*\| \to 0$ in probability as $k \to \infty$, where $\theta^*$ denotes a solution to (3). Additionally, $\psi_k$ converges weakly (in 2-Wasserstein distance) to the distribution $\pi_{\mathcal{N}}(\psi|\theta^*)$. Consequently, the algorithm enables proper uncertainty quantification for $\psi$-related quantities, such as the $V$-function and $Q$-value function, which are central to RL. Notably, uncertainty quantification for value-functions is generally beyond the reach of conventional iterative optimization algorithms used to train actor-critic models. With this latent trajectory formulation, we establish an essentially trajectory-independent training framework for actor-critic models.

## 3.2 Adaptive Stochastic Gradient MCMC for Deep Actor-Critic Learning

To perform $\psi$-sampling using SGMCMC, we need to evaluate the gradient $\nabla_\psi \log \pi_{\mathcal{N}}(\psi_k|\theta_{k-1})$. This can be done using the following identity established in Song et al. (2020):

$$\nabla_\psi \log \pi(\psi|\theta) = \int \nabla_\psi \log \pi(\psi|\boldsymbol{z}, \theta) \pi(\boldsymbol{z}|\psi, \theta) d\boldsymbol{z},$$

where $\boldsymbol{z}$ denotes a latent variable. By treating trajectory observation $(\boldsymbol{x}^{(k)}, \boldsymbol{R}^{(k)})$ as latent variables, we can derive the following formula:

$$\nabla_\psi \log \pi_{\mathcal{N}}(\psi_k|\theta_{k-1})$$

$$= \int \nabla_\psi \log \pi_{\mathcal{N}}(\psi_k|\boldsymbol{x}^{(k)}, \boldsymbol{R}^{(k)}, \theta_{k-1}) \pi(\boldsymbol{x}^{(k)}, \boldsymbol{R}^{(k)}|\psi_k, \theta_{k-1}) d\boldsymbol{x}^{(k)} d\boldsymbol{R}^{(k)}$$

$$= \int \nabla_\psi \log \pi_{\mathcal{N}}(\psi_k|\boldsymbol{x}^{(k)}, \boldsymbol{R}^{(k)}) \frac{\pi(\boldsymbol{x}^{(k)}, \boldsymbol{R}^{(k)}|\psi_k, \theta_{k-1})}{\pi(\boldsymbol{x}^{(k)}, \boldsymbol{R}^{(k)}|\theta_{k-1})} \pi(\boldsymbol{x}^{(k)}, \boldsymbol{R}^{(k)}|\theta_{k-1}) d\boldsymbol{x}^{(k)} d\boldsymbol{R}^{(k)}$$

$$= \int \nabla_\psi \log \pi_{\mathcal{N}}(\psi_k|\boldsymbol{x}^{(k)}, \boldsymbol{R}^{(k)}) \frac{\pi(\boldsymbol{R}^{(k)}|\boldsymbol{x}^{(k)}, \psi_k, \theta_{k-1}) \pi(\boldsymbol{x}^{(k)}|\psi_k, \theta_{k-1})}{\pi(\boldsymbol{R}^{(k)}|\boldsymbol{x}^{(k)}, \theta_{k-1}) \pi(\boldsymbol{x}^{(k)}|\theta_{k-1})} \pi(\boldsymbol{x}^{(k)}, \boldsymbol{R}^{(k)}|\theta_{k-1}) d\boldsymbol{x}^{(k)} d\boldsymbol{R}^{(k)}$$

$$= \int \nabla_\psi \log \pi_{\mathcal{N}}(\psi_k|\boldsymbol{x}^{(k)}, \boldsymbol{R}^{(k)}) \frac{\pi(\boldsymbol{R}^{(k)}|\boldsymbol{x}^{(k)}, \psi_k)}{\pi(\boldsymbol{R}^{(k)}|\boldsymbol{x}^{(k)})} \pi(\boldsymbol{x}^{(k)}, \boldsymbol{R}^{(k)}|\theta_{k-1}) d\boldsymbol{x}^{(k)} d\boldsymbol{R}^{(k)}$$

$$\tag{4}$$

provided that the mini-batch size $n$ has been chosen to be sufficiently large, ensuring that $\boldsymbol{x}^{(k)}$ serves as a good representative of the underlying pseudo-population. This guarantees that $\psi_k$ is conditionally independent of $\theta_{k-1}$ given $\boldsymbol{x}^{(k)}$. Consequently, we have $\nabla_\psi \log \pi_{\mathcal{N}}(\psi_k|\boldsymbol{x}^{(k)}, \boldsymbol{R}^{(k)}, \theta_{k-1}) = \nabla_\psi \log \pi_{\mathcal{N}}(\psi_k|\boldsymbol{x}^{(k)}, \boldsymbol{R}^{(k)})$ and $\pi(\psi_k|\boldsymbol{x}^{(k)}) = \pi(\psi_k|\theta_{k-1})$. The former leads to the second equality in (4), while the latter leads to the last equality in (4) by the following equality:

$$\frac{\pi(\boldsymbol{x}^{(k)}|\psi_k, \theta_{k-1})}{\pi(\boldsymbol{x}^{(k)}|\theta_{k-1})} = \frac{\pi(\psi_k|\boldsymbol{x}^{(k)}, \theta_{k-1})}{\pi(\psi_k|\theta_{k-1})} = \frac{\pi(\psi_k|\boldsymbol{x}^{(k)})}{\pi(\psi_k|\theta_{k-1})} = 1. \tag{5}$$

To facilitate evaluation of the likelihood function $\pi(\boldsymbol{R}^{(k)}|\boldsymbol{x}^{(k)}, \psi_k)$, we made the following assumption regarding the distribution of $R_t$:

**Assumption 1** *The conditional distribution $\pi(R_t|x_t, \psi)$ is Gaussian and defined explicitly as:*

$$R_t|x_t, \psi \sim \mathcal{N}(V_\psi(s_t), \sigma^2). \tag{6}$$

Note that the Gaussian assumption for the reward has also been made under the Kalman Temporal Difference framework, see e.g. Geist & Pietquin (2010), Tripp & Shachter (2013), and Shashua & Mannor (2020).

**Remark 1** *How to evaluate $\nabla_\psi \log \pi_\mathcal{N}(\psi_k | \boldsymbol{x}^{(k)}, \boldsymbol{R}^{(k)})$? Based on Assumption 1, we have*

$$\nabla_\psi \log \pi_\mathcal{N}(\psi_k | \boldsymbol{x}^{(k)}, \boldsymbol{R}^{(k)}) = \nabla_\psi \log \pi_\mathcal{N}(\boldsymbol{R}^{(k)} | \boldsymbol{x}^{(k)}, \psi_k) + \nabla_\psi \log \pi(\psi_k)$$

$$= \frac{\mathcal{N}}{n} \sum_{i=1}^n \nabla_\psi \log \pi(R_i^{(k)} | x_i^{(k)}, \psi_k) + \nabla_\psi \log \pi(\psi_k)$$

*where $\pi(\psi_k)$ denotes the prior distribution of $\psi_k$.*

**Remark 2** *How to evaluate the importance weight $w_k = \frac{\pi(\boldsymbol{R}^{(k)} | \boldsymbol{x}^{(k)}, \psi_k)}{\pi(\boldsymbol{R}^{(k)} | \boldsymbol{x}^{(k)})}$? Since the numerator can be evaluated based on Assumption 1, we consider the evaluation of the denominator in this remark. One way is to evaluate the denominator based on the relationship:*

$$\pi(\boldsymbol{R}^{(k)} | \boldsymbol{x}^{(k)}) = \int \pi(\boldsymbol{R}^{(k)} | \boldsymbol{x}^{(k)}, \psi_k) \pi(\psi_k | \boldsymbol{x}^{(k)}) d\psi_k, \tag{7}$$

*i.e., estimating the denominator by averaging the density $\pi(\boldsymbol{R}^{(k)} | \boldsymbol{x}^{(k)}, \psi_k)$ over a set of samples of $\psi_k$ drawn from $\pi(\psi_k | \boldsymbol{x}^{(k)})$. The auxiliary samples of $\psi_k$ can be simulated using a SGMCMC algorithm based on the following gradient:*

$$\nabla_{\tilde{\psi}} \log \pi(\tilde{\psi} | \boldsymbol{x}^{(k)}) = \int \nabla_{\tilde{\psi}} \log \pi(\tilde{\psi} | \boldsymbol{x}^{(k)}, \tilde{\boldsymbol{R}}) \pi(\tilde{\boldsymbol{R}} | \tilde{\psi}, \boldsymbol{x}^{(k)}) d\tilde{\boldsymbol{R}}, \tag{8}$$

*which can be estimated based on auxiliary samples of $\tilde{\boldsymbol{R}}$ drawn from $\pi(\tilde{\boldsymbol{R}} | \tilde{\psi}, \boldsymbol{x}^{(k)})$, as defined in Assumption 1.*

*Alternatively, one can estimate $\pi(\boldsymbol{R}^{(k)} | \boldsymbol{x}^{(k)})$ using the Nadaraya-Watson (NW) conditional density kernel estimator:*

$$\hat{\pi}(R | x) = \frac{\sum_{i=1}^n K_{h_2}(x - x_i^{(k)}) K_{h_1}(R - R_i^{(k)})}{\sum_{i=1}^n K_{h_2}(x - x_i^{(k)})}, \tag{9}$$

*where both $K_{h_1}(\cdot)$ and $K_{h_2}(\cdot)$ are Gaussian kernels, and $h_1$ and $h_2$ are their respective bandwidths. The NW estimator is known to be consistent provided $h_1 \to 0$, $h_2 \to 0$, and $nh_1h_2 \to \infty$ as $n \to \infty$ (Hyndman et al., 1996). Extensions of the NW estimator based on local polynomial smoothing are available, see e.g., Fan et al. (1996) and Gooijer & Zerom (2003). See Izbicki & Lee (2016) for an estimator in a high-dimensional regression setting.*

As a summary, we have Algorithm 1, which provides an efficient implementation for the proposed Latent Trajectory Framework. Although the algorithm is described to perform a single update of $\psi$ at each sampling step, multiple updates are also allowed. This does not interfere with the convergence theory of the algorithm.

### 3.3 CONVERGENCE THEORY

The LTF training process is essentially an adaptive SGMCMC algorithm. In this framework, we simulate $\psi_k \sim \pi(\psi_k | \theta_{k-1})$ with SGMCMC algorithms while $\theta_{k-1}$ changes from iteration to iteration. We establish the convergence theory for parameters both actor network $\pi_\theta$ and critic network $V_\psi$, as detailed in Algorithm 1. We prove the $L_2$-convergence of $\theta_k$ and the $\mathcal{W}_2$-convergence of $\psi_k$. This implies that the actor network achieves an optimal policy and the critic network converges weakly to the stationary distribution $\pi(\psi | \theta^*)$.

**Theorem 3.1 (Convergence of $\theta_k$)** *Suppose Assumptions 2-6 hold, and the sample size of auxiliary $\tilde{\psi}$-samples is sufficiently large. If we set the learning rate sequence $\{\epsilon_k\}_{k=1}^\infty$ and the step size sequence $\{\omega_k\}_{k=1}^\infty$ as the form:*

$$\epsilon_k = \frac{C_\epsilon}{c_\epsilon + k^\alpha}, \quad \omega_k = \frac{C_\omega}{c_\omega + k^\beta}, \tag{15}$$

*then there exists a root $\theta^* \in \{\theta : g(\theta) = 0\}$ such that*

$$\mathbb{E}\|\theta_k - \theta^*\|^2 \le \xi \omega_k, \quad k \ge k_0, \tag{16}$$

*for some constant $\xi > 0$ and iteration number $k_0 > 0$.*

---

**Algorithm 1** Latent Trajectory Framework for A2C (LT-A2C)

---

1: Initialize actor network $\pi_{\theta_0}$ with learning rate sequence $\{\omega_k\}$
2: Initialize critic network $V_{\psi_0}$ with learning rate sequence $\{\epsilon_k\}$
3: **for** $k = 1, \ldots, \mathcal{K}$ **do**
4:     Generate trajectories $\boldsymbol{x}^{(k)} = \{x_i^{(k)}\}_{i=1}^n$ and returns $\boldsymbol{R}^{(k)} = \{R_i^{(k)}\}_{i=1}^n$ with policy $\pi_{\theta_{k-1}}$
5:     **Step 1: Draw auxiliary samples of** $\tilde{\psi}$
6:     **for** $j = 1, \ldots, m$ **do**
7:         **Presetting:** If $j = 1$, set $\tilde{\psi}_0 = \psi_{k-1}$

$$\tilde{\psi}_j = \tilde{\psi}_{j-1} + \frac{\delta_j}{2} \nabla_{\tilde{\psi}} \log \pi(\tilde{\psi}|\boldsymbol{x}^{(k)}) + \tilde{e}_j, \tag{10}$$

where $\nabla_{\tilde{\psi}} \log \pi(\tilde{\psi}|\boldsymbol{x}^{(k)}) = \frac{1}{L} \sum_{i=1}^L \nabla_{\tilde{\psi}} \log \pi(\tilde{\psi}|\boldsymbol{x}^{(k)}, \tilde{\boldsymbol{R}}_i)$ is calculated based on (8) using $L$ auxiliary samples of $\tilde{\boldsymbol{R}}$ drawn from $\pi(\tilde{\boldsymbol{R}}|\tilde{\psi}, \boldsymbol{x}^{(k)})$, $\delta_j$ is the learning rate, and $\tilde{e}_j \sim N_p(0, \delta_j I_p)$.
8:     **end for**
9:     **Step 2: Sampling** $\psi_k$ **through SGMCMC**
10:     **Importance weight:** calculate

$$\hat{w}_k = \frac{\pi(\boldsymbol{R}^{(k)}|\boldsymbol{x}^{(k)}, \psi_{k-1})}{\frac{1}{m} \sum_{j=1}^m \pi(\boldsymbol{R}^{(k)}|\boldsymbol{x}^{(k)}, \tilde{\psi}_j)}. \tag{11}$$

11:     **Sampling:** Draw $e_k \sim N_p(0, \frac{n}{\mathcal{N}} \epsilon_k I_p)$ and calculate

$$\psi_k = \psi_{k-1} + \frac{\epsilon_k}{2} \nabla_\psi \tilde{L}(\theta_{k-1}, \psi_{k-1}) + e_k, \tag{12}$$

where the gradient term is given by

$$\nabla_\psi \tilde{L}(\theta_{k-1}, \psi_{k-1}) = \hat{w}_k \Big\{ \sum_{i=1}^n \nabla_\psi \log \pi(R_i^{(k)}|x_i^{(k)}, \psi_{k-1}) + \frac{n}{\mathcal{N}} \nabla_\psi \log \pi(\psi_k) \Big\}. \tag{13}$$

12: **end for**
13: **Step 3: Updating** $\theta_k$ **through SGD**
14: Compute advantage function $A_{\psi_k}(x_i^{(k)}, R_i^{(k)})$ with equation (1).

$$\theta_k = \theta_{k-1} + \omega_k \sum_{i=1}^n A_{\psi_k}(x_i^{(k)}, R_i^{(k)}) \nabla_\theta \log \pi_{\theta_{k-1}}(a_i^{(k)}|s_i^{(k)}). \tag{14}$$

---

Let $\pi^* = \pi(\psi|\theta^*)$, let $T_k = \sum_{i=0}^{k-1} \epsilon_{i+1}$, and let $\mu_{T_k}$ denote the probability law of $\psi_k$. Theorem 3.2 establishes convergence of $\mu_{T_k}$ in 2-Wasserstein distance.

**Theorem 3.2 ($\mathcal{W}_2$-convergence of $\psi_k$)** *Suppose Assumptions 2-7 hold, the sample size of auxiliary $\tilde{\psi}$-samples is sufficiently large, and $\{\epsilon_k\}$ and $\{\omega_k\}$ are set as in Theorem 3.1. Then, for any $k \in \mathbb{N}$,*

$$\mathcal{W}_2(\mu_{T_k}, \pi^*) \le (\hat{C}_0 \delta_{\tilde{L}}^{1/4} + \tilde{C}_1 \gamma_1^{1/4}) T_k + \hat{C}_2 e^{-T_k/c_{LS}},$$

*for some positive constants $\hat{C}_0$, $\hat{C}_1$, and $\hat{C}_2$, where $\mathcal{W}_2(\cdot, \cdot)$ denotes the 2-Wasserstein distance, $c_{LS}$ denotes the logarithmic Sobolev constant of $\pi^*$, and $\delta_{\tilde{L}}$ is a coefficient as defined in Assumption 3 and reflects the variation of the stochastic gradient $\nabla_\psi \tilde{L}(\theta_{k-1}, \psi_k)$.*

We prove Theorems 3.1 and 3.2 by following the proof of adaptive SGLD in Liang et al. (2024). We note that the proposed latent trajectory framework can also be implemented using adaptive SGHMC (Liang et al., 2022b). In this case, Theorems 3.1 and 3.2 can still be established similar to the convergence theory presented in Liang et al. (2022b).

## 4 EXPERIMENTS

In this section, we evaluate the performance and effectiveness of the LTF in enhancing actor-critic algorithms. We conduct experiments in two environments: the simple Escape Environment, where we demonstrate the ability of uncertainty quantification for LTF-enhanced algorithms, and the Py-Bullet Environment (Ellenberger, 2018–2019), where we compare the performance of LT-A2C to vanilla A2C on continuous control benchmarks. These experiments highlight the improvements in training stability and performance metrics achieved through the adoption of LTF.

### 4.1 UNCERTAINTY QUANTIFICATION

In this simple experiment, we demonstrate the ability of the proposed method in uncertainty quantification for the Actor-Critic network using the Escape Environment designed in (Shih & Liang, 2024). Figure 2 depicts a simple escape environment, for which the state space consists of 100 grids and the agent's objective is to navigate to the goal positioned at the top right corner. The agent starts its task from the bottom left grid at time $t = 0$. For every time step $t$, the agent identifies its current position, represented by the coordinate $s = (x, y)$. Given a policy $\pi_\theta$, the agent chooses an action $a \in \{N, E, S, W\}$ with respect to the probability $\pi_\theta(a|s)$. The action taken by the agent determines the adjacent grid to which it moves. Following each action, the agent is awarded an immediate reward, $r_t$, drawn from the Gaussian distribution $\mathcal{N}(-1, 0.01)$.

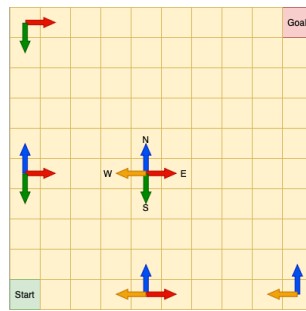

Figure 2: Indoor escape environment

We evaluate the performance of the proposed method from three aspects: (i) Policy Diversity: The policy, coded by the actor network and denoted as $\pi_\theta(a|s)$, should converge to a distribution that assigns equal probabilities to optimal actions and zero probability to others. (ii) Value Accuracy: The critic network is expected to accurately approximate the state value function $V^*(s)$ across the entire state space. (iii) Value Uncertainty: Algorithms must be capable of quantifying the uncertainty associated with value function.

To quantify policy diversity, we define the optimal policy distribution $\pi^*(\cdot|s)$ be a probability distribution over all actions at state $s$, where $\pi^*(\cdot|s)$ is uniform on optimal actions and zero on sub-optimal actions. For a given policy $\pi_\theta(\cdot|s)$, the Kullback-Leibler Divergence (KL Divergence) between $\pi^*$ and $\pi_\theta$, denoted by $D_{KL}(\pi^*\|\pi_\theta)$, can be used to measure the diversity of the policy distribution. It's worth noting that for most states, actions N and E are identically optimal. Hence, the policy $\pi_\theta(a|s)$ should assign the same probability on these two actions. Figure 3 and Figure 4 visualize the policy probability of $\pi_\theta$ at each state $s$. The left figure demonstrates a policy with a small $D_{KL}(\pi^*\|\pi_\theta)$, While, the right figure shows a policy with a large $D_{KL}(\pi^*\|\pi_\theta)$.

Suppose the actor network converges to a fixed policy $\pi_{\theta^*}$, and the state value function $V_\psi(s)$ coded by the critic network should be distributed around the optimal value function $V^*(s)$. To evaluate

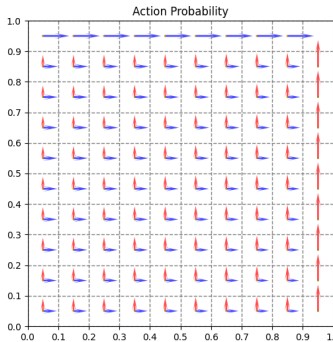
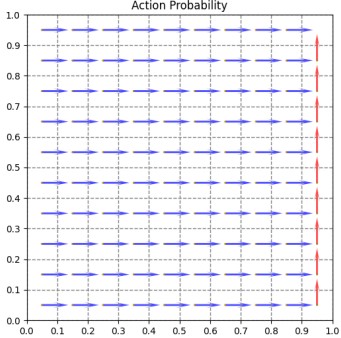

Figure 3: $D_{KL}(\pi^*\|\pi_\theta)$ small      Figure 4: $D_{KL}(\pi^*\|\pi_\theta)$ large

such estimation, we collect the last 1000 parameter updates to form a $\psi$-sample pool, denoted by $\boldsymbol{\psi}_s = \{\hat{\psi}_i\}$, which naturally induces a sample pool of value functions $\mathbf{V}_s = \{V_\psi(\cdot)|\psi \in \boldsymbol{\psi}_s\}$. From $V$-sample pool, We can obtain a point estimate of the state value at $s$ by calculating the sample average $\hat{V}(s) = \frac{1}{n}\sum_{i=1}^n V_{\hat{\psi}_i}(s, a)$. For interval estimation, we can achieve one-step value tracking by constructing a 95% prediction interval with the state value samples. We replicate each experiment 100 times and calculate the following three metrics: (1) the KL-divergence between $\pi^*$ and $\pi_{\theta^*}$, denoted by $D_{KL}(\pi^*\|\pi_{\theta^*})$, (2) the mean squared error (MSE) between $\hat{V}(s)$ and $V^*(s)$, defined by $\mathrm{MSE}(\hat{V}) = \mathbb{E}_{s\in\mathcal{S}}(\hat{V}(s) - V^*(s))^2$, where $\mathbb{E}(\cdot)$ denotes the empirical average over the state space $\mathcal{S}$; and (3) the coverage rate (CR) of the 95% prediction intervals.

In Table 1, we demonstrate that with the aid of the LTF, the A2C and PPO algorithms exhibit significant improvements in all three performance indicators. We further visualize the experimental results in Figures 5 and 6, where LTF-enhanced algorithms, LT-A2C and LT-PPO achieve smaller $\mathrm{MSE}(\hat{V})$ for point estimation and a nearly 95% coverage rate for interval estimation. Moreover, due to the correct estimation of $V^*$, $\pi_{\theta^*}$ converges to an approximately optimal policy distribution. In contrast, vanilla A2C and PPO algorithms suffer from significant bias in the value function estimation. The proposed LTF method effectively addresses this inconsistency, providing consistent estimates and reliable uncertainty quantification for the value function.

From a computational perspective, LTF is essentially an optimization step combined with a Bayesian sampling loop, where the sampling is performed using SGMCMC algorithms. LTF can be easily scaled up to large neural networks, enhancing its applicability.

Table 1: Metrics for Escape Environment

| Algorithm | $\mathcal{N}$ | $D_{KL}(\pi^*\|\pi_{\theta^*})$ | $\mathrm{MSE}(\hat{V})$ | Coverage Rate | CI-Width |
|---|---|---|---|---|---|
| A2C | - | 4.647 (0.0729) | 0.53527 (0.03974) | 0.489 (0.0061) | 0.413 (0.0023) |
| LT-A2C | 10000 | 0.010 (0.0010) | 0.00038 (0.00001) | 0.947 (0.0004) | 0.457 (0.0009) |
| LT-A2C | 20000 | 0.014 (0.0014) | 0.00039 (0.00001) | 0.947 (0.0004) | 0.452 (0.0010) |
| LT-A2C | 40000 | 0.014 (0.0013) | 0.00033 (0.00001) | 0.947 (0.0004) | 0.449 (0.0009) |
| PPO | - | 4.773 (0.0893) | 0.56112 (0.04272) | 0.487 (0.0066) | 0.416 (0.0024) |
| LT-PPO | 10000 | 0.011 (0.0010) | 0.00041 (0.00001) | 0.947 (0.0004) | 0.458 (0.0009) |
| LT-PPO | 20000 | 0.009 (0.0009) | 0.00038 (0.00001) | 0.947 (0.0005) | 0.452 (0.0009) |
| LT-PPO | 40000 | 0.011 (0.0011) | 0.00032 (0.00001) | 0.947 (0.0004) | 0.449 (0.0008) |

## 4.2 PYBULLET ENVIRONMENT

To demonstrate the applicability of the LTF-enhanced actor-critic algorithms in complex environments, we evaluate the performance of the LT-A2C algorithm on continuous control benchmarks using the RL Baselines3 Zoo Raffin (2020) and the PyBullet environment Ellenberger (2018–2019).

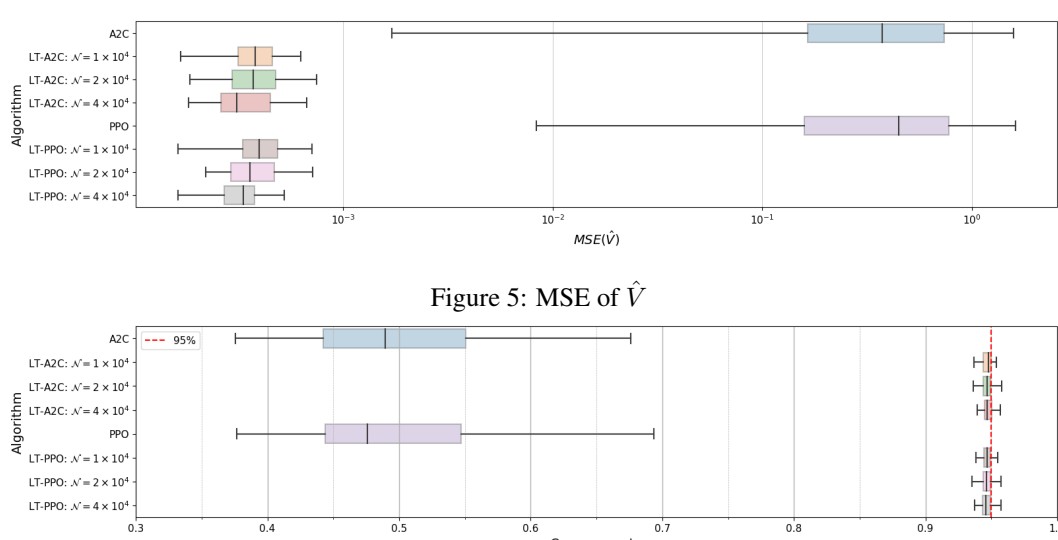

Figure 5: MSE of $\hat{V}$

Figure 6: Coverage rate of the 95% prediction interval of the value $V^*(s)$

We compare LT-A2C to vanilla A2C across four tasks: HalfCheetah, Hopper, Reacher, and Walker. The hyperparameters for A2C follow the settings in RL Baselines3 Zoo Raffin (2020). Further experimental settings are detailed in the Appendix.

Figure 7 presents three key metrics for evaluating algorithm performance: (i) training reward, (ii) evaluation reward, and (iii) the best model reward up to time t. Each experiment was replicated 50 times to assess overall performance. The colored bands in the figure represent the variability in rewards at each training step, providing insight into the robustness of the algorithms. Narrower bands indicate lower variability, suggesting greater resistance to random initialization and environmental stochasticity. The results demonstrate that LT-A2C consistently achieves higher training rewards compared to vanilla A2C, while also exhibiting reduced variability. This highlights the effectiveness of the Latent Trajectory Framework in improving policy exploration while maintaining robustness.

As shown in Algorithm 1, LT-A2C and traditional A2C share the same policy control mechanism, specifically the use of the policy gradient. The key difference lies in the policy evaluation step, where LT-A2C incorporates SGMCMC sampling. In previous experiments, we demonstrated that the LT framework enhances the ability to quantify uncertainties in the value function, leading to improved policy exploration and control. Due to the flexibility of the LT framework, it can be applied to a wide range of existing algorithms to boost their performance by refining policy evaluation. By integrating the latent trajectory framework, the RL model not only improves training efficiency but also reduces variability, resulting in more robust and reliable performance across diverse environments.

## 5 CONCLUSION

In this paper, we introduce a novel latent trajectory framework as an alternative to conventional optimization methods for deep actor-critic models and propose an adaptive SGMCMC algorithm for training these models. We rigorously prove that, under mild conditions, the proposed algorithm guarantees consistency in parameter estimation for the actor network and weak convergence in parameter sampling for the critic network. A key advantage of our method is its ability to accurately quantify the uncertainty of the value functions central to reinforcement learning, while simultaneously improving performance in practical applications. Importantly, our framework achieves these improvements without sacrificing computational complexity, making it scalable to large-scale neural networks. The framework is also highly flexible: beyond replacing the SGLD steps with SGHMC (Chen et al., 2014) or substituting the SGD step with Adam (Kingma & Ba, 2014), we can also replace the policy control step with a PPO update scheme. This flexibility enables seamless integration with advanced machine learning techniques, further enhancing the framework's potential to

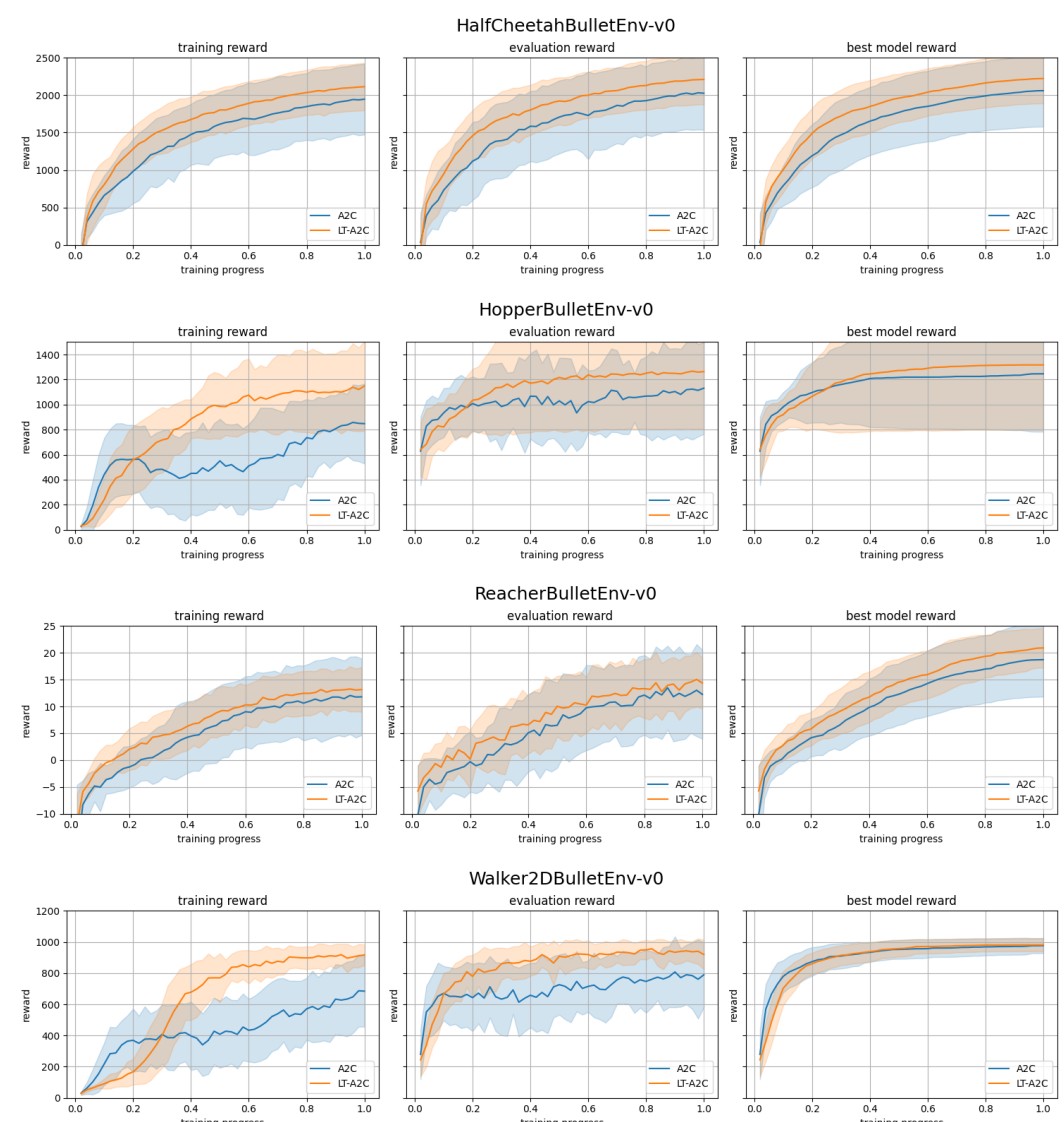

Figure 7: PyBullet training performance

improve performance while maintaining robust uncertainty quantification, making it a powerful tool for large-scale reinforcement learning tasks.

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

## A  PROOF OF THEOREM 3.1 AND THEOREM 3.2

For convenience, we denote the trajectory observation $(\boldsymbol{x}^{(k)}, \boldsymbol{R}^{(k)})$ as $\boldsymbol{z}_k$, and assume $\boldsymbol{z}_k \in \mathcal{Z}$ be a compact set. The Latent Trajectory Framework can be written in a general form as

$$
\begin{aligned}
\psi_k &= \psi_{k-1} + \epsilon_k \nabla_\psi \tilde{L}(\theta_{k-1}, \psi_{k-1}, \boldsymbol{z}_k) + \sqrt{2\epsilon_k} e_k, \\
\theta_k &= \theta_{k-1} + \omega_k \tilde{g}(\theta_{k-1}, \psi_k, \boldsymbol{z}_k),
\end{aligned}
\tag{17}
$$

where $\epsilon_k$ denotes the learning rate, $e_k$ is a standard Gaussian noise, $\nabla_\psi \tilde{L}(\theta_{k-1}, \psi_{k-1}, \boldsymbol{z}_k)$ denotes an unbiased estimate of $\nabla_\psi L(\theta_{k-1}, \psi_{k-1}) = \nabla_\psi \log \pi(\psi_{k-1}|\theta_{k-1})$, and $\tilde{g}(\theta_{k-1}, \psi_k, \boldsymbol{z}_k)$ is an unbiased estimator of $g_{\psi_k}^{ac}(\theta_{k-1})$. Convergence of adaptive stochastic gradient MCMC algorithms has been studied in Deng et al. (2019), Dong et al. (2023) and Liang et al. (2024). The convergence theory of LTF can be established by slightly modifying some of the assumptions.

**Notation:** We use $\mathbb{E}_\psi[u(\theta, \psi)]$ to denote the expectation of $u(\theta, \psi)$ with respect to the conditional distribution $\pi(\psi|\theta)$, and use $\mathbb{E}[u(\cdot)]$ to denote the expectation with respect to the joint distribution of all the variables involved in the integrand $u(\cdot)$.

**Assumption 2** *The step size sequence $\{\omega_k\}_{k\in\mathbb{N}}$ is a positive decreasing sequence of real numbers such that*

$$
\lim_{k\to\infty} \omega_k = 0, \quad \sum_{k=1}^\infty \omega_k = \infty.
\tag{18}
$$

*There exist $\delta > 0$ and a stationary point $\theta^*$ such that for any $\theta \in \Theta$,*

$$
\langle \theta - \theta^*, g(\theta) \rangle \leq -\delta \|\theta - \theta^*\|^2,
$$

*where $g(\theta) = \mathbb{E}_\psi[g_\psi^{ac}(\theta)]$ and, in addition,*

$$
\liminf_{k\to\infty} 2\delta \frac{\omega_k}{\omega_{k+1}} + \frac{\omega_{k+1} - \omega_k}{\omega_{k+1}^2} > 0,
\tag{19}
$$

*where $\|\cdot\|$ denotes the $L^2$-norm.*

**Assumption 3** *$L(\theta, \psi)$ is $M$-smooth on $\theta$ and $\psi$ with $M > 0$, and $(m, b)$-dissipative on $\psi$ for some constants $m > 1$ and $b > 0$. In other words, for any $\psi, \psi', \psi'' \in \Psi$ and $\theta, \theta' \in \Theta$, the following inequalities are satisfied:*

$$
\|\nabla_\psi L(\theta, \psi') - \nabla_\psi L(\theta', \psi'')\| \leq M\|\psi' - \psi''\| + M\|\theta - \theta'\|,
\tag{20}
$$

$$
\langle \nabla_\psi L(\theta^*, \psi), \psi \rangle \leq b - m\|\psi\|^2,
\tag{21}
$$

*where $\theta^*$ is a stationary point as defined in Assumption 2.*

Assumption 2 is a critical and standard assumption in the convergence of SGMCMC algorithms. In the context of deep neural networks, the dissipativity condition can be easily achieved by imposing a Gaussian prior on the critic network parameter, which further guarantees convergence.

**Lemma A.1** *$\|\nabla_\psi L(\theta, \psi)\|^2 \leq 3M^2\|\psi\|^2 + 3M^2\|\theta - \theta^*\|^2 + 3B^2$ for some constant $B$.*

PROOF: Follow the proof of Lemma A.1 in Dong et al. (2023). □

**Assumption 4** *Let $\zeta_k = \nabla_\psi \tilde{L}(\theta_k, \psi_k, \boldsymbol{z}_k) - \nabla_\psi L(\theta_k, \psi_k)$. Assume that $\zeta_k$'s are mutually independent white noises, and they satisfy the conditions*

$$
\mathbb{E}(\zeta_k|\mathcal{F}_k) = 0, \quad \mathbb{E}\|\zeta_k\|^2 \leq \delta_{\tilde{L}}(M^2\mathbb{E}\|\psi_k\|^2 + M^2\mathbb{E}\|\theta_k - \theta^*\|^2 + B^2),
\tag{22}
$$

*where $\delta_{\tilde{L}}$ and $B$ are positive constants, and $\mathcal{F}_k = \sigma\{\theta_1, \psi_1, \theta_2, \psi_2, \ldots, \theta_k, \psi_k\}$ denotes a $\sigma$-filtration.*

**Assumption 5** *There exist positive constants $M$ and $B$ such that for all $\boldsymbol{z} \in \mathcal{Z}$,*

$$
\|\tilde{g}(\theta, \psi, \boldsymbol{z})\| \leq M^2\|\theta - \theta^*\|^2 + M^2\|\psi\|^2 + B^2,
$$

*where $\tilde{g}(\theta, \psi, \boldsymbol{z})$ is as defined in (17).*

By the formulation defined in section 3.2, let $g(\theta) = \mathbb{E}_{(\psi,z)}[\tilde{g}(\theta, \psi, z)|\theta]$ and $\eta = \tilde{g}(\theta, \psi, z) - g(\theta)$. Since $\mathbb{E}_{(\psi,z)}[\|\tilde{g}(\theta, \psi, z)\|^2|\theta] = \|g(\theta)\|^2 + \mathbb{E}_{(\psi,z)}[\|\eta\|^2|\theta]$, this implies $\mathbb{E}\|g(\theta)\|^2 \leq \mathbb{E}\|\tilde{g}(\theta, \psi, z)\|^2$ and $\mathbb{E}\|\eta\|^2 \leq \mathbb{E}\|\tilde{g}(\theta, \psi, z)\|^2$.

**Lemma A.2** *(Uniform $L^2$ bounds) Suppose Assumptions 2-5 hold. If the following conditions are satisfied:*

$$\epsilon_k = \frac{C_\epsilon}{c_\epsilon + k^\alpha}, \quad \omega_k = \frac{C_\omega}{c_\omega + k^\beta}, \tag{23}$$

*for some constants $C_\epsilon > 0$, $c_\epsilon > 0$, $C_\omega > 0$, $c_\omega > 0$, $\alpha, \beta \in (0, 1]$, and $\beta \leq \alpha \leq \min\{1, 2\beta\}$. Then there exist constants $G_\psi$ and $G_\theta$ such that $\mathbb{E}\|\psi_k\|^2 \leq G_\psi$ and $\mathbb{E}\|\theta_k - \theta^*\|^2 \leq G_\theta$ for all $k = 0, 1, 2, \dots$.*

PROOF: Follow the proof of Lemma A.2 in Dong et al. (2023). We slightly modify Assumption 4 in Dong et al. (2023) by Assumption 5, where the stochastic gradient is replaced with $\hat{g}_\psi^{ac}(\theta)$. Then the proof is straight forward.

□

**Assumption 6** *(Solution of Poisson equation) For any $\theta \in \Theta$, $\psi \in \Psi$, and a function $\mathcal{V}(\psi) = 1 + \|\psi\|$, there exists a function $\mu_\theta$ on $\Psi$ that solves the Poisson equation $\mu_\theta(\psi) - \mathcal{T}_\theta \mu_\theta(\psi) = g_\psi^{ac}(\theta) - g(\theta)$, where $\mathcal{T}_\theta$ denotes a probability transition kernel with $\mathcal{T}_\theta \mu_\theta(\psi) = \int_\Psi \mu_\theta(\psi') \mathcal{T}_\theta(\psi, \psi') d\psi'$, such that*

$$g_{\psi_{k+1}}^{ac}(\theta_k) = g(\theta_k) + \mu_{\theta_k}(\psi_{k+1}) - \mathcal{T}_{\theta_k} \mu_{\theta_k}(\psi_{k+1}), \quad k = 1, 2, \dots. \tag{24}$$

*Moreover, for all $\theta, \theta' \in \Theta$ and $\psi \in \Psi$, we have $\|\mu_\theta(\psi) - \mu_{\theta'}(\psi)\| + \|\mathcal{T}_\theta \mu_\theta(\psi) - \mathcal{T}_{\theta'} \mu_{\theta'}(\psi)\| \leq \varsigma_1 \|\theta - \theta'\| \mathcal{V}(\psi)$ and $\|\mu_\theta(\psi)\| + \|\mathcal{T}_\theta \mu_\theta(\psi)\| \leq \varsigma_2 \mathcal{V}(\psi)$ for some constants $\varsigma_1 > 0$ and $\varsigma_2 > 0$.*

PROOF OF THEOREM 3.1

PROOF: For Algorithm 1, we assume that the sample size of auxiliary $\tilde{\psi}$-samples is sufficiently large, ensuring the denominator estimator in Eq. (11) converges almost surely to its mean value (Teh et al., 2016). Therefore, the resulting stochastic gradient (13) is almost surely unbiased.

Dong et al. (2023) proved the result (16) for a more general adaptive Langevinized ensemble Kalman filter (LEnKF) algorithm, which is equivalent to an adaptive pre-conditioned SGLD algorithm. Extending their proof to Algorithm 1 is straight forward. □

**Assumption 7** *The probability law $\mu_0$ of the initial hypothesis $\theta_0$ has a bounded and strictly positive density $p_0$ with respect to the Lebesgue measure on $\mathbb{R}_{d_\psi}$, and*

$$\kappa_0 := \log \int_{\mathbb{R}^{d_\psi}} e^{\|\theta\|^2} p_0(\theta) d\theta < \infty.$$

PROOF OF THEOREM 3.2

PROOF: This theorem is proved in Liang et al. (2024) with the same Assumptions 2-7. For Algorithm 1, we only need to assume that the sample size $m$ of auxiliary $\tilde{\psi}$-samples is sufficiently large, as explained in the proof of Theorem 3.1. □

# B EXPERIMENT SETTINGS

## B.1 ESCAPE ENVIRONMENT

In this experiment, both $\pi_\theta$ and $V_\psi$ are approximated by deep neural networks with two hidden layers of sizes (128, 128). The agent updates the network parameters every 50 interactions, for a total of $10^6$ action steps. Each experiment is replicated for 100 times. For initial exploration, an entropy penalty coefficient of 0.01 is added, and gradually decay to 0. To achieve sparse deep neural network, we follow the suggestion in Sun et al. (2022) to impose mixture Gaussian prior onto both network parameters:

$$\theta, \psi \sim (1 - \lambda)\mathcal{N}(0, \sigma_0^2) + \lambda\mathcal{N}(0, \sigma_1^2) \tag{25}$$

where $\lambda \in (0, 1)$ is the mixture proportion and $\sigma_0^2$ is usually set to a small number compare to $\sigma_1^2$. We set $\sigma_1 = 0.01$, $\sigma_0 = 0.001$ and $\lambda = 0.5$ in all LTF-enhanced algorithms. For indoor escape environment, the reward is given by $\mathcal{N}(-1, 0.01)$; that is, we set $\sigma^2 = 0.01$. To make the estimated return $y_t = R_t$ stationary, the reward at the goal state is set to $\mathcal{N}(-1, \frac{0.01}{1-\gamma^2})$, where the discount factor $\gamma = 0.9$. To guarantee the convergence of LTF, we set the decay policy learning rate as $\omega_k = O(\frac{1}{k^{0.5}})$ and constant critic learning rate $\epsilon_k = 2 \times 10^{-4}$. The sample size $L$ in (10) is set to 50, and the auxiliary sample size $m$ in (11) is set to 5.

In practical implementation, drawing samples from the conditional distribution $\pi_\mathcal{N}(\psi_k|\boldsymbol{x}_k)$ can be performed with a short sub-loop of SGMCMC updates, which we set the length to be 10. That is, for each iteration $k$, we repeat the $\psi$-sampling update 10 times. The sub-loop sampling scheme is given by

$$\psi_{k,\ell} = \psi_{k,\ell-1} + \frac{\epsilon_{k,\ell}}{2}\hat{w}_{k,\ell}\Big\{ \sum_{i=1}^{n} \nabla_\psi \log \pi(R_i^{(k)}|x_i^{(k)}, \psi_{k,\ell-1}) + \frac{n}{\mathcal{N}}\nabla_\psi \log \pi(\psi_{k,\ell-1}) \Big\} + e_{k,\ell} \tag{26}$$

where the sub-loop is indexed by $\ell$. And the importance weight can be calculated by

$$\hat{w}_{k,\ell} = \frac{\pi(\boldsymbol{R}^{(k)}|\boldsymbol{x}^{(k)}, \psi_{k,\ell-1})}{\frac{1}{m+1}\sum_{j=1}^{m} \pi(\boldsymbol{R}^{(k)}|\boldsymbol{x}^{(k)}, \tilde{\psi}_j) + \frac{1}{m+1}\pi(\boldsymbol{R}^{(k)}|\boldsymbol{x}^{(k)}, \psi_{k,\ell-1})}.$$

where $m$ denote the number of auxiliary samples and the importance weight is bounded by $m + 1$. The boundedness of the importance weights $\hat{w}_{k,\ell}$'s further ensures the stability of SGMCMC sampling step. We note that including the $\psi_{k,l-1}$-term in the denominator is reasonable. As implied by the definition of the importance weight $w_k = \frac{\pi(\boldsymbol{R}^{(k)}|\boldsymbol{x}^{(k)}, \psi_k)}{\pi(\boldsymbol{R}^{(k)}|\boldsymbol{x}^{(k)})}$, the numerator term should be part of the denominator and, therefore, we need to include $\psi_{k,l-1}$ as an auxiliary sample of $\tilde{\psi}$. Furthermore, we refer to Theorem 1 of Song et al. (2020) for the sample equally weighted formula in calculating the denominator.

In Figure 8, we present boxplots of four metrics for each algorithm. Across all metrics, LT-A2C and LT-PPO outperform the traditional A2C and PPO algorithms, demonstrating lower MSE, lower KL-Divergence, and higher coverage rates. Lower KL-divergence indicates that the policy distribution converges to a uniform distribution over optimal actions, leading to more efficient exploration and robust learning. For $MSE(\hat{V})$, LTF-enhanced algorithms have significantly smaller values and tighter boxplots, indicating training stability. Regarding uncertainty quantification, only LTF algorithms achieve the desired 95% coverage rate. As the pseudo population increases, more accurate uncertainty quantification for both point and interval estimates is achieved, as evidenced by smaller MSE and narrower interval ranges.

Regarding computation complexity, although LTF requires additional SGMCMC sampling on critic network parameter in each iteration, the complexity for SGLD and SGHMC are the same as stochastic gradient methods. Therefore, the total time complexity remains that same, which implies the scalablility of proposed framework.

## B.2 PYBULLET ENVIRONMENT

In this experiment, we conduct experiments on PyBullet environments, including Ant, HalfCheetah, Hopper, Reacher, and Walker2D. The training framework and hyperparameters of A2C are based

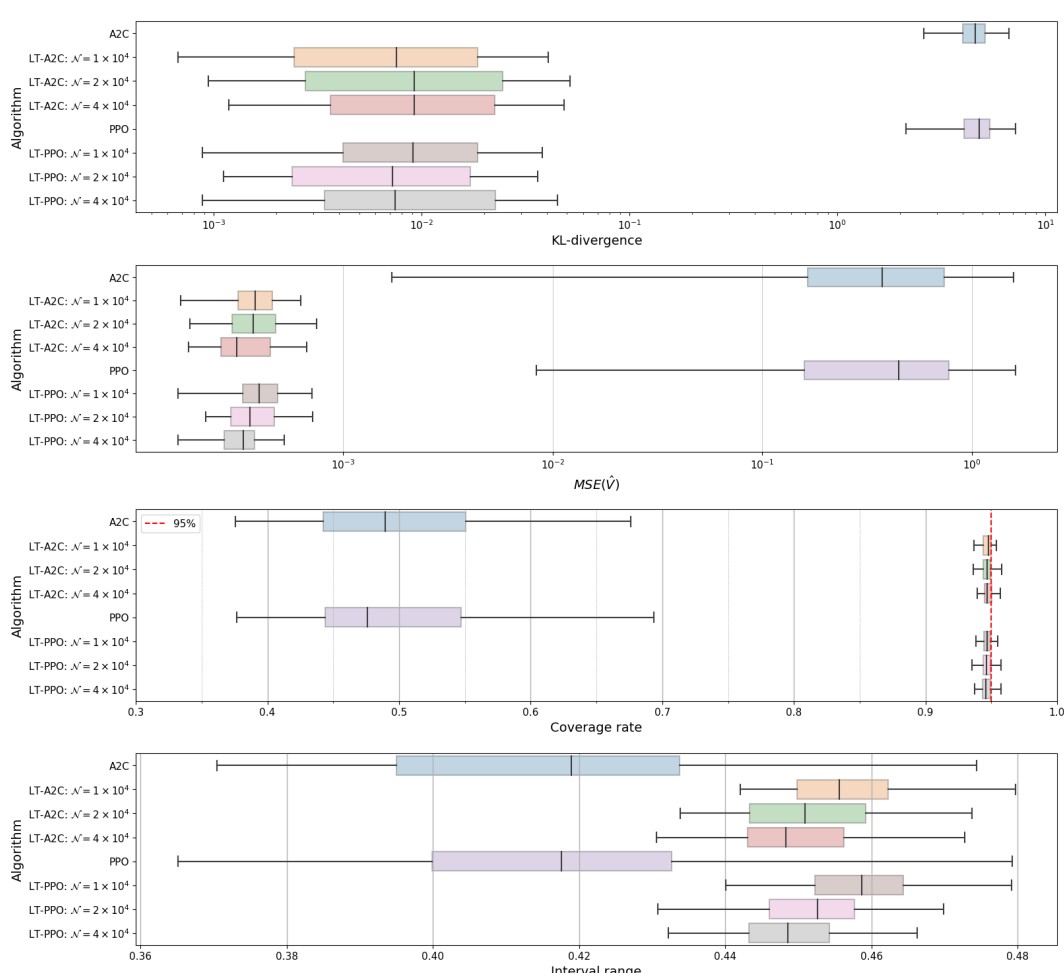

Figure 8: Metrics for Escape environment

on RL baselines3 zoo, and our LT-A2C is implemented on top of Stable-Baselines3 Raffin et al. (2021). The hyperparameters are given in Table 2 and 3. Actor and critic network, both have 2 hidden layers of size [64, 64]. There 3 types of learning rate, constant, linear decay and polynomial decay. To balance between exploration and exploitation in LT-A2C, we adopt an annealing technique, where the pseudo population size increases as training steps increase, starting from 500. This method allows the algorithm to gradually shift from exploration to exploitation, improving overall performance and stability. A2C algorithm optimize both network with RMSprop, and LT-A2C update actor network with RMSprop as well. The A2C algorithm optimizes both the actor and critic networks using the RMSprop optimizer. LT-A2C also updates the actor network with RMSprop, while using SGHMC for critic parameter sampling. For the prior distribution, LT-A2C employs the same Gaussian mixture prior as used in the Escape environment.

In theory, the auxiliary sampling step requires large sample size to guarantee a good approximation. To improve the sampling efficiency of the auxiliary sampling step, we modify the approximation procedure of the importance weight. We replace the auxiliary samples $\psi_{k,j}$'s with the SGMCMC samples $\psi_{k,\ell}$ derived in (26). The importance weight can then be approximated by

$$\hat{w}_{k,\tilde{\ell}} = \frac{\pi(\boldsymbol{R}^{(k)}|\boldsymbol{x}^{(k)}, \psi_{k,\tilde{\ell}-1})}{\frac{1}{\tilde{\ell}} \sum_{\ell=0}^{\tilde{\ell}-1} \pi(\boldsymbol{R}^{(k)}|\boldsymbol{x}^{(k)}, \psi_{\ell})}.$$

With this modification, we can eliminate the auxiliary sampling step and further lower the computation complexity and memory complexity.

Table 2: Hyperparameters

| Environment | HalfCheetah | | Hopper | |
|---|---|---|---|---|
| Hyperparameters | LT-A2C | A2C | LT-A2C | A2C |
| learning rate | lin 0.00067 | lin 0.00096 | lin 0.00042 | lin 0.00096 |
| $\sigma$ (observation) | 0.1 | - | 0.1 | - |
| $\mathcal{N}$ | 50000 | - | 10000 | - |
| $\gamma$(discount factor) | 0.95 | 0.99 | 0.99 | 0.99 |
| gae-$\lambda$ | 0.9 | 0.9 | 1.0 | 0.9 |
| train batch | 32 | 32 | 32 | 32 |
| training steps | 2e6 | 2e6 | 2e6 | 2e6 |

Table 3: Hyperparameters (cont.)

| Environment | Reacher | | Walker2D | |
|---|---|---|---|---|
| Hyperparameters | LT-A2C | A2C | LT-A2C | A2C |
| learning rate | lin 0.00096 | lin 0.0008 | lin 0.00037 | lin 0.00096 |
| $\sigma$ (observation) | 0.1 | - | 0.1 | - |
| $\mathcal{N}$ | 1000 | - | 500 | - |
| $\gamma$(discount factor) | 0.99 | 0.99 | 0.99 | 0.99 |
| gae-$\lambda$ | 1.0 | 0.9 | 1.0 | 0.9 |
| train batch | 32 | 32 | 32 | 32 |
| training steps | 2e6 | 2e6 | 2e6 | 2e6 |

