# OpenReview forum: "Latent Trajectory: A New Framework for Actor-Critic Reinforcement Learning with Uncertainty Quantification"
_ICLR.cc/2025/Conference — Submitted to ICLR 2025_

### Official Review · Reviewer_8ewQ · 2024-10-29

**Soundness:** 1
**Presentation:** 1
**Contribution:** 2
**Rating:** 3
**Confidence:** 2

**Summary:**

This paper introduces a Latent Trajectory Framework (LTF) to improve uncertainty quantification in deep actor-critic reinforcement learning, addressing the challenge of value function uncertainty in stochastic environments. Using an adaptive Stochastic Gradient Markov Chain Monte Carlo (SGMCMC) algorithm, the method enables trajectory-independent training, backed by theoretical convergence guarantees and empirical performance improvements. This approach enhances both the robustness and reliability of RL applications by integrating latent transition trajectories.

**Strengths:**

Theoretical Analysis: The paper provides a theoretical analysis of the Latent Trajectory Framework, attempting to establish convergence and benefits for uncertainty quantification in RL.

**Weaknesses:**

**$.1**
- L47: Why is the actor considered "unknown"? We can access its weights during training, so we should be able to evaluate it on any state-action pair. Even if the actor were unknown, how does that affect uncertainty quantification?
General: Could you clarify the type of uncertainty you’re addressing?

**$.2**
- L100: What is π(x∣θ)? I thought π represented the policy, a distribution over the action space.
- L103: What is π(ψ∣θ)?
- L105: What does "pseudo-population size" mean? Is N not equal to the batch size n?
- L107: Similar to above, this line is unclear.
- Eq(3): Could you provide the intuition behind this learning objective?

**$.3**
- How does this approach differ from the SGMCMC method discussed in Shih & Liang (2024)?

**$.4**
- $4.1: The writing lacks organization. For instance, metrics are introduced at L323, but the computation details are only explained two paragraphs later. Why is coverage rate the chosen metric for uncertainty quantification?
- $4.2: Since actor-critic methods are typically used for continuous action spaces, why not use Mujoco benchmarks for Fig. 7? Additionally, can LT be extended to SAC or other recent methods?

**Overall**
- This paper heavily relies on prior work for explanations and notations, which makes it challenging for readers unfamiliar with the domain to follow.

**Questions:**

none

---

> ### Author Response · Authors · 2024-11-23
> **Reply to Reviewer 8ewQ**
>
> **[W1]**
> >- Thank you for pointing out the unclear term "unknown". We will change the term "unknown actor network" to "non-fixed actor network", where the existing uncertainty quantification algorithms can only deal with one neural network. When it comes to actor-critic algorithms, the dependency between two network parameter update will introduce additional complexity in uncertainty quantification.
> We are aware of the types of uncertainty: aleatory uncertainty and epistemic uncertainty.
> Since we are working on the uncertainty of the V-function, which is an expectation of the cumulated rewards, the uncertainty we considered is epistemic uncertainty.
>
> **[W2]**
> >- For a given actor network parameter $\theta$, the policy $\pi_\theta$ is fixed. The stationary distribution of transition tuples is uniquely determined by the environment and the policy, hence, we assume that we can drawn transition tuples independently from the stationary distribution $\pi(x|\theta)$.
>
> >- Given a policy $\pi_\theta$, the value function $V_\psi$ is learned to approximate its true value $V^{\pi_\theta}$, therefore, the network weights $\psi$ can be viewed as a random variable conditioned on $\theta$.
>   In our paper, $\pi(\psi|\theta)$ represents ``the conditional distribution of $\psi$ given $\theta$''.
>
> >- In the setting of online RL, we have access to infinite data from the interaction between the agent and the environment. Like traditional Bayesian inference, if the sample size is infinity, the posterior distribution will degenerate to a point. Hence, the role of the "pseudo-population size" is to prevent the degeneration of the conditional distribution $\pi(\psi|\theta)$, which is denoted by $\pi_\mathcal{N}(\psi|\theta)$.
>
> >- Eq(3) integrates out the critic network parameter $\psi$ to mimic the policy gradient objective, while taking account of the uncertainty of $\psi$ conditioned on $\theta$. The uncertainty within $\pi_\mathcal{N}(\psi|\theta)$ comes from the transition generating process and $\psi$ updating process.
>
> **[W3]**
> >- Shih \& Liang (2024) focuses on DQN type methods, which only involves one deep neural network. We extend the SGMCMC framework to actor-critic methods, which has to consider the convergence of both neural networks. Since two network parameters affect each other during the training, this extension is highly non-trivial.
>
> **[W4]**
> >- The coverage rate is generally regarded as the most rigorous metric for uncertainty quantification because it directly evaluates the alignment between predicted uncertainty and actual outcomes. It rigorously tests calibration, ensures reliability in probabilistic predictions, and detects overconfidence or underconfidence in models.
>
> >- The PyBullet environment also features a continuous action space and serves as a more implementation-friendly alternative to MuJoCo. The latent trajectory (LT) framework can be seamlessly integrated into existing algorithms, such as PPO, by simply replacing the critic optimization step with our SGMCMC sampling step.
> Moreover, the LT framework can perform uncertainty quantification even after an existing algorithm has converged (post-training inference). This is possible because the LT framework does not require any changes to the network architecture or the introduction of additional parameters, ensuring compatibility and ease of adoption.

---

### Official Review · Reviewer_y7uj · 2024-10-31

**Soundness:** 2
**Presentation:** 2
**Contribution:** 2
**Rating:** 3
**Confidence:** 4

**Summary:**

The authors propose a novel method for characterising uncertainty in value functions using a stochastic gradient MCMC algorithm. They carry out a convergence analysis of their method before evaluating PyBullet and Gridworld environments.

**Strengths:**

The theoretical analysis of the proposed algorithm seems sounds from a cursory readying. Convergence guarantees are always welcome in papers.

**Weaknesses:**

My main concern relates to lack of positioning of the paper. Bayesian RL offers a precise way to characterise uncertainty in an MDP. At every timestep, the posterior over uncertain variables updates according to a Bayesian Bellman operator. Uncertainty can be characterised in the state-reward transitions as in model-based approaches or other sufficient variables like value functions or Bellman operators as in model-free approaches[4].  There is already a wealth of literature in uncertainty quantification in value functions. See my question below to authors.

The authors claim `Notably, uncertainty quantification for value-functions is generally beyond the reach of conventional iterative optimization algorithms used to train actor-critic models'. This is not true. Methods such optimistic actor-critic [1], BBAC[2] and EVE[3] (to name but a few) have been able to quantify uncertainty in value functions when used in continuous control for some time now as uncertainty quantification is essential to their exploration methods. There also exist analyses of the various approximate inference tools used to quantify uncertainty [5]. See [6] for a recent comparison of state of the art continuous control using uncertainty quantification evaluated in a variety of domains.

Empirical weaknesses:

There is a significant lack of comparison to other methods that quantify uncertainty in value functions. A comparison of these methods seems essential to evaluate the contribution of the proposed method. Moreover, the authors don't indicate number of timesteps in their evaluations so it is difficult to gauge the worth of their approach in comparison to similar Bayesian methods.

[1] Coisek et al., Better Exploration with Optimistic Actor-Critic, 2019, https://arxiv.org/pdf/1910.12807
[2] Fellows et al., Bayesian Bellman Operators, 2023, https://arxiv.org/pdf/2106.05012
[3] Schmitt et al., Exploration via Epistemic Value Estimation, 2023, https://arxiv.org/pdf/2303.04012
[4] Fellows et al., Bayesian Exploration Networks, 2024 https://arxiv.org/pdf/2308.13049
[5] Coisek et al., Conservative Uncertainty Estimation By Fitting Prior Networks, 2020 https://openreview.net/pdf?id=BJlahxHYDS
[6]  Tasdighi et al., Deep Exploration with PAC-Bayes, 2025, https://arxiv.org/pdf/2402.03055

**Questions:**

How does the proposed method fit into the general model-free Bayesian RL framework that precisely characterises uncertainty in value functions? How does $\pi_\mathcal{N}(\psi\vert \theta)$ relate to the posterior over $g_\psi$ under these frameworks? If it differs from existing characterisations, ie [1]-[4], what theoretical, algorithmic or empirical advantages does it offer over a full Bayesian approach for characterising uncertainty in value functions?

Can the authors approach be used to derived Bayes-optimal policies? If not, why does their uncertainty quantification prevent this?

Can the authors extend their empirical evaluation to include other methods that quantify uncertainty in their value functions?

---

> ### Author Response · Authors · 2024-11-23
> **Reply to Reviewer y7uj**
>
> **General reply**
> Thank you for providing extensive references, which we have thoroughly reviewed. From our understanding, the uncertainty of value functions must be rigorously quantified in a **distributional sense**. However, as highlighted in our meta-reply, the uncertainty quantification methods in these papers lack rigorous validation. Specifically, none of the referenced works have verified that the “uncertainty” they claim to measure is accurate. For instance, no evidence is provided to confirm that the confidence intervals they produce achieve the desired coverage rates.
>
> Additionally, these Bayesian methods fail to fully account for the dynamics of the RL system. For example, in Algorithm 1 of the Bayesian Bellman Operator (BBO) paper, samples are continuously generated in a dynamic system but are incorrectly treated as identically distributed. Bayesian methods aim to introduce uncertainty into the MAP estimates by using random priors, which can lead to randomized estimates. However, there are no theoretical guarantees for posterior consistency or for the reliability of downstream inference based on these estimates. The BBO paper adopts a similar strategy, further highlighting this limitation.
>
> **[Q1]** How does the proposed method fit into the general model-free Bayesian RL framework that precisely characterises uncertainty in value functions? How does relate to the posterior over under these frameworks? If it differs from existing characterizations, ie [1]-[4], what theoretical, algorithmic or empirical advantages does it offer over a full Bayesian approach for characterizing uncertainty in value functions?
>
> **[A1]** In Algorithm 1 (LT-A2C), as shown in Eq. (14), the actor network update remains identical to the A2C algorithm. The only modification lies in the critic network sampling step. Notably, the LT framework does not require any changes to the network architecture to achieve uncertainty quantification. Similarly, by replacing the critic network update step in PPO, we can extend the framework to obtain LT-PPO.
>
> The advantages of our approach and the limitations of existing Bayesian methods are thoroughly discussed in the meta reply. Additionally, Bayesian methods often face significant challenges when extending their theoretical results to deep neural network settings. These methods typically rely on linear approximation assumptions, which restrict their scalability and practical applicability.
>
> In more complex settings, Bayesian methods frequently resort to using approximated MAP estimates instead of sampling algorithms. This compromise sacrifices their ability to perform uncertainty quantification, undermining a key advantage of Bayesian approaches.
>
> **[Q2]** Can the authors approach be used to derived Bayes-optimal policies? If not, why does their uncertainty quantification prevent this?
>
> **[A2]** We can derive Bayesian version of our algorithm by adding prior information on the actor network parameter, and this will not affect the performance of the proposed algorithm in uncertainty quantification. Also, we would like to point out that we proved the convergence (in probability) for the policy network and the weak convergence of the actor network. Adding a prior on the actor network will not change this theoretical network.
>
> **[Q3]**  Can the authors extend their empirical evaluation to include other methods that quantify uncertainty in their value functions?
>
> **[A3]** We will include comparisons with other actor-critic methods in the uncertainty quantification (UQ) experiments. However, it is important to note that these existing algorithms are typically based on Q-learning methods combined with an actor network, and they do not treat the dynamic system as an integrated whole. In (Shih & Liang, 2024), it was demonstrated that existing Bayesian Q-learning methods struggle to accurately estimate Q-functions and construct reliable confidence intervals. We will make sure to address this limitation in the related works section of our revised manuscript.

---

### Official Review · Reviewer_5DS5 · 2024-11-03

**Soundness:** 2
**Presentation:** 3
**Contribution:** 2
**Rating:** 5
**Confidence:** 3

**Summary:**

Uncertainty quantification of the value function is crucial for a robust reinforcement learning algorithm. This work considers the challenging actor-critic setting. The introduced latent trajectory framework (LTF) is built upon the adaptive Stochastic Gradient Markov chain Monte Carlo (SGMCMC) by treating the transition trajectory and the value function parameter as latent variables for policy optimization. The proposed method is theoretically proved to be able to converge under mild conditions. The experiments on indoor escape environments  and PyBullet environment show that the proposed method has better performance compared with baseline A2C algorithms.

**Strengths:**

- Theoretical results: The proposed LTF is grounded by theoretical convergence results. The paper contains most of the proof details, and the logic in the writing is easy to follow.
- Experiments: The experiments show multiple metrics for evaluating the performance of the proposed LTF. For instance, the KL and MSE can help evaluate the performance from different perspectives. The results in HalfCheetah show that the proposed method can effectively improve the performance compared with A2C.

**Weaknesses:**

- The proposed LTF only compares with the vanilla A2C method, while there have been many  AC-based methods proposed, such as [W1,W2]. I recognize that the proposed LTF introduces a new perspective by treating the value parameters as a latent variable, while the experiments are lacking. In particular, the experiments in Section 4.1 are conducted on rather new environments (by Liang et al.). The lack of comparison with other AC methods (e.g., in experiments and/or related works section) makes the results and performance gain less convincing. If the comparison is not possible or not necessary, please clarify the reasons.
- The proposed LTF is largely based on SGMCMC algorithms that are proposed in Liang et al., 2022a; Deng et al.,2019, while the major contribution of the LTF is less clear. From my understanding, the main contribution is on the A2C settings, which pose unique challenges for uncertainty quantification. It will be very beneficial for the authors to explicitly state the key challenges of applying SGMCMC to A2C settings and how their approach addresses these challenges.




[W1] Zhou et al. "Natural actor-critic for robust reinforcement learning with function approximation." Advances in neural information processing systems 36 (2024).

[W2] Wu, et al. "Uncertainty weighted actor-critic for offline reinforcement learning." arXiv preprint arXiv:2105.08140 (2021).

**Questions:**

- What is the parameter $\delta_j$ in Eqn. 10?
- What are the relation between $\epsilon_{k,l}$ and $\epsilon_{k} $ in Algorithm 1
- How to select $\mathcal{N}$ in practice?
- Why the variance in Figure 7 for LT-A2C is not significantly reduced if the uncertainty is effectively evaluated during update (e.g., HopperBullet, Evaluation reward)?

---

> ### Author Response · Authors · 2024-11-23
> **Reply to Reviewer 5DS5**
>
> **[W1]** The proposed LTF only compares with the vanilla A2C method, while there have been many AC-based methods proposed, such as [W1,W2]. I recognize that the proposed LTF introduces a new perspective by treating the value parameters as a latent variable, while the experiments are lacking. In particular, the experiments in Section 4.1 are conducted on rather new environments (by Liang et al.). The lack of comparison with other AC methods (e.g., in experiments and/or related works section) makes the results and performance gain less convincing. If the comparison is not possible or not necessary, please clarify the reasons.
>
>
> **response:** In [W1], the algorithm is built on linear assumptions, while our algorithm leverages a deep critic network, enabling it to model nonlinear dynamics and adapt to more sophisticated tasks effectively.
> In [W2], the algorithm is specifically designed for offline RL, which operates under the assumption of a fixed dataset. This is fundamentally different from our online RL setting, where data is dynamically generated during training. These differing assumptions make direct comparisons using the same baseline challenging and less meaningful.
>
> **[W2]** The proposed LTF is largely based on SGMCMC algorithms that are proposed in Liang et al., 2022a; Deng et al.,2019, while the major contribution of the LTF is less clear. From my understanding, the main contribution is on the A2C settings, which pose unique challenges for uncertainty quantification. It will be very beneficial for the authors to explicitly state the key challenges of applying SGMCMC to A2C settings and how their approach addresses these challenges.
>
> **response:** First of all, formulating RL as an adaptive SGMCMC problem is highly non-trivial, which needs to conceptualize the transition tuples as latent variables. Consequently, this leads to an ideal **transition-trajectory independent** algorithm for RL. We united the 2-step optimization problem (policy control and policy evaluation) to solving a single integral equation (eq.(3)).
>
> Secondly, applying the adaptive SGMCMC algorithm to the A2C setting
> is highly challenging. Specifically, the gradient evaluation for
> $\nabla_{\psi_k} \log \pi_{\mathcal{N}}(\psi_k|\theta_{k-1})$ is highly
> non-trivial, which involves an importance weighting step and simulation of
> auxiliary samples.
>
> **[Q1]** What is the parameter in Eqn. 10?
> **[A1]** Equation (10) is used to draw auxiliary samples of $\tilde{\psi}$, and  $\tilde{\psi}$ can be understood as its parameter.
>
>
> **[Q2]** What are the relation between $\epsilon_{k,l}$ and $\epsilon_k$ in Algorithm 1?
> **[A2]**  Sorry, it is typo: $\epsilon_{k,l}$ needs to be replaced with $\epsilon_k$.
>
>
> **[Q3]** How to select $\mathcal{N}$ in practice?
> **[A3]** This is not a critical issue for the proposed algorithm. It can be chosen as a reasonably large number, ensuring the resulting confidence interval to be reasonably narrow as the simulation goes on.
>
>
> **[Q4]**  Why the variance in Figure 7 for LT-A2C is not significantly reduced if the uncertainty is effectively evaluated during update (e.g., HopperBullet, Evaluation reward)?
> **[A4]** The variance of the reward can be influenced by various factors, including update frequency, mini-batch size, and environment complexity, among others. The primary focus of this experiment is to demonstrate that incorporating the Latent Trajectory (LT) framework does not negatively impact the practical performance of the actor-critic algorithm. Importantly, the LT framework achieves accurate uncertainty quantification without compromising the algorithm’s effectiveness or efficiency.

---

### Official Review · Reviewer_pDCR · 2024-11-04

**Soundness:** 3
**Presentation:** 2
**Contribution:** 2
**Rating:** 3
**Confidence:** 2

**Summary:**

This paper introduces the latent trajectory framework (LTF) that implicitly models the uncertainty of Q-functions by drawing multiple samples of critic parameters, essentially forming a distribution over Q-values.

**Strengths:**

- The paper provides theroretical justification for the convergence of the proposed method.
- The insight of conditional independence between the critic parameters and past actor parameters given the current state trajectory is particularly interesting.

**Weaknesses:**

## Lack of related works and comparison
It is hard to position this paper in the context of prior work as it lacks a discussion of how this work relates to existing RL algorithms that model the uncertainty of Q-values. The paper should discuss how this work is different from distributional RL methods, which also model the uncertainty of Q-values. The paper should also discuss why SGMCMC is the suitable method for the problem at hand, in contrast to prior work. This made the paper particularly hard to understand as a reader.

## The problem of uncertainty quantification is not well-motivated
The paper does not provide a clear motivation for why it is important to model the uncertainty of Q-values. It mentions "accurately quantifying the uncertainty of the value function has been a critical concern for ensuring reliable and robust RL applications", but does not provide any concrete examples of why this is the case or precisely in what scenarios this is important. Ideally, the paper should analyze the limitations of existing RL algorithms that do not model the uncertainty of Q-values and provide examples of scenarios where this leads to suboptimal performance, and how the proposed method addresses these limitations in experiments.

## No results on PPO
The paper mentions that PPO suffers from severe miscalibration issues in both the actor and critic, but does not provide any results on PPO to demonstrate this.

## Connecting uncertainty quantification to performance
While the escape environment discusses the relationship of how LTF leads to better MSE of value functions, how this translates to better performance in the escape environment is not clear. Conversely, on other environments, the paper does not provide a clear analysis of how the uncertainty quantification of Q-values leads to better performance.

It is also not so clearly apparent that LT-A2C has smaller seed variability than A2C, as the paper claims. The performance difference is largely imperceptible in the plots, considering the confidence intervals.

**Questions:**

Can you build a connection to prior works in uncertainty quantification?

How would you make this paper more approachable to a wider audience? Currently, it requires knowing a lot of prior work to make sense of the motivation, algorithm, and convergence proofs. The paper should be largely self-sufficient when reading.

---

> ### Author Response · Authors · 2024-11-23
> **Reply to Review pDCR**
>
> **[W1] Lack of related works and comparison.** It is hard to position this paper in the context of prior work as it lacks a discussion of how this work relates to existing RL algorithms that model the uncertainty of Q-values. The paper should discuss how this work is different from distributional RL methods, which also model the uncertainty of Q-values. The paper should also discuss why SGMCMC is the suitable method for the problem at hand, in contrast to prior work. This made the paper particularly hard to understand as a reader.
> **response:** Refer to our common reply regarding Bayesian RL. Bayesian RL and distributional RL (e.g., QR-DQN) address different aspects of uncertainty: Bayesian RL quantifies the uncertainty in Q-networks, whereas QR-DQN focuses on quantifying the uncertainty of rewards. In this paper, we focus exclusively on quantifying value function uncertainties; therefore, distributional RL falls outside the scope of this work. Furthermore, as demonstrated in Shih & Liang (2024), QR-DQN fails to construct accurate confidence intervals for true Q-values. Conversely, Bayesian methods heavily rely on techniques such as bootstrapping or Gaussian processes (suitable primarily for linear representations), which are challenging to extend effectively to deep neural networks both theoretically and numerically.
>
> The proposed **adaptive** SGMCMC algorithm is particularly suitable
> for RL because of the following facts:
> >- Comprehensive Handling of RL Dynamics:
> The proposed algorithm fully accounts for the dynamic nature of reinforcement learning. At each time step, new transition tuples are generated based on the updated policy, leading to a continuously evolving distribution of transition tuples. Consequently, the target distribution for the critics in the SGMCMC method also changes at every time step due to updates in the actor network. We have proven the convergence of the actor network, which in turn ensures the convergence of the critic network’s distribution.
>
> >- Independence from Transition Trajectories:
> Our algorithm treats transition tuples as latent (or auxiliary) variables, integrating them out when updating the actor network. This approach guarantees that the convergence of the actor network is independent of the transition trajectory. In contrast, many existing reinforcement learning methods, such as Bayesian RL, rely heavily on trajectory-dependent computations.
>
> >- No Additional Complexity or Structural Changes:
> The proposed algorithm does not require modifications to the network architecture and maintains the same computational complexity as existing methods. This simplicity makes it easily applicable to large-scale problems and real-world applications.
>
> **[W2] The problem of uncertainty quantification is not well-motivated.**
> The paper does not provide a clear motivation for why it is important to model the uncertainty of Q-values. It mentions "accurately quantifying the uncertainty of the value function has been a critical concern for ensuring reliable and robust RL applications", but does not provide any concrete examples of why this is the case or precisely in what scenarios this is important. Ideally, the paper should analyze the limitations of existing RL algorithms that do not model the uncertainty of Q-values and provide examples of scenarios where this leads to suboptimal performance, and how the proposed method addresses these limitations in experiments.
>
> **response:** Thank you for your suggestion. The importance of  uncertainty quantification has been well addressed in existing works like distributional RL and Bayesian RL. However, the "uncertainty" is not rigorously defined and verified. In this paper, we provide a much more effective method than the existing ones in uncertainty quantification; specifically, our method results in highly accurate coverage rates for value function.
>
> **[W3] No results on PPO**
> The paper mentions that PPO suffers from severe miscalibration issues in both the actor and critic, but does not provide any results on PPO to demonstrate this.
> **response:** Sorry for the confusion, we intend to state that the PPO leads to less accurate coverage rates and higher prediction error for value function. This has been demonstrated in Table 1 and Figure 6.

---

> > ### Comment · Reviewer_pDCR · 2024-12-02
> >
> > Thank you for your response, but my concerns remain largely unaddressed.
> > > In this paper, we focus exclusively on quantifying value function uncertainties; therefore, distributional RL falls outside the scope of this work.
> > I disagree with this, because the motivation is the same (the authors themselves agree with this when they say "importance of uncertainty quantification has been well addressed in existing works like distributional RL and Bayesian RL"). Therefore, the methods also apply to the problem this paper addresses, albeit might require minimal modifications. There should be better baselines in the paper to give a fair analysis of what the prior work is capable of when addressing the problem proposed in this work. That would make the empirical section of this paper significantly stronger.
> >
> > > While performance improvements are not the primary focus, our results underscore that accurate uncertainty estimation can be achieved without trade-offs in algorithm efficiency or effectiveness.
> >
> > This is in disagreement with the claims of the paper "We provide theoretical guarantees for the convergence of our algorithm and offer empirical evidence showing improvements in both performance and robustness of the deep actor-critic model under our Latent Trajectory Framework (LTF)."
> >
> > I do believe performance improvements should be possible, but they need to be demonstrated clearly empirically. It does not have to be all environments, but there should be clear evidence of when UQ would help and when it would match the baseline performance.

---

> > > ### Author Response · Authors · 2024-12-03
> > >
> > > Thank you for highlighting the issues related to the numerical experiments. We would like to address your concerns as follows:
> > >
> > > First, we have included the coverage rate of QR-DQN (distributional RL) in Table 1 of the meta reply. This demonstrates that QR-DQN fails to provide correct interval estimates for the true Q-values. The primary reason is that constructing accurate interval estimates requires a theoretical guarantee of the weak convergence of network parameters. Unfortunately, existing methods lack this guarantee under the deep neural network setting, which undermines their reliability for uncertainty quantification.
> > >
> > > Second, regarding your concern about how uncertainty quantification (UQ) improves existing algorithms when adopting the LT framework, we conducted additional experiments. Specifically, we evaluated the success rate of learning optimal policies in the “Indoor Escape Environment” over 100 independent runs. As shown in the attached figure, LT-A2C achieves a 100% success rate without relying on random exploration or entropy penalty, whereas Vanilla A2C achieves less than 80%. This demonstrates the performance improvements of adopting LT-framework.
> > >
> > > https://pasteboard.co/SM2eLIQizSap.png

---

> ### Author Response · Authors · 2024-11-23
> **Reply to Review pDCR (continue)**
>
> **[W4] Connecting uncertainty quantification to performance.** While the escape environment discusses the relationship of how LTF leads to better MSE of value functions, how this translates to better performance in the escape environment is not clear. Conversely, on other environments, the paper does not provide a clear analysis of how the uncertainty quantification of Q-values leads to better performance.
>
> **response:** Uncertainty quantification (UQ) is crucial for understanding the entire RL system. In the indoor escape environment, we observe that an accurate value function approximation, indicated by a low mean squared error (MSE), facilitates more diverse action exploration across optimal policies (see Figure 3). In contrast, bias in value function estimation negatively impacts the policy control step, causing the policy network to converge prematurely to a local minimum without adequately exploring other optimal policies, as illustrated in Figure 4. In the PyBullet experiments, the only distinction between A2C and LT-A2C lies in the policy evaluation step. The performance improvements achieved by LT-A2C highlight the advantage of accurate policy evaluation, enabled by the Latent Trajectory framework.
>
> **[W5] It is also not so clearly apparent that LT-A2C has smaller seed variability than A2C, as the paper claims. The performance difference is largely imperceptible in the plots, considering the confidence intervals.**
>
> **response:** Our two experiments focus on different aspects of the proposed Latent Trajectory (LT) framework:
> >- Theoretical Validation in the Indoor Escape Environment:
> In this experiment, we demonstrate the theoretical rigor of the LT framework. Specifically, we show that LT-A2C achieves the expected coverage rate of 95\% for the confidence intervals of the value function $V^*(s)$. This validates that our framework provides reliable uncertainty quantification, a critical requirement for theoretical soundness.
>
> >- Performance Improvement in Pybullet:
> The second experiment focuses on demonstrating the practical performance of the Latent Trajectory (LT) framework. By integrating the LT framework into an actor-critic algorithm and evaluating it on a suite of Pybullet tasks, we show that the framework maintains or slightly improves policy performance compared to standard actor-critic methods.
>
> The key takeaway is that the LT framework achieves correct uncertainty quantification without compromising performance. While performance improvements are not the primary focus, our results underscore that accurate uncertainty estimation can be achieved without trade-offs in algorithm efficiency or effectiveness.
>
> **[Q1]** Can you build a connection to prior works in uncertainty quantification?
> **response:** The connections with Bayesian RL and distributional RL have been established in our replies to your comment W1 and the meta reply. We will add the related work to our revision shortly.
>
> **[Q2]** How would you make this paper more approachable to a wider audience? Currently, it requires knowing a lot of prior work to make sense of the motivation, algorithm, and convergence proofs. The paper should be largely self-sufficient when reading.
> **response:**  Due to the page limit, we will incorporate your comments into the revision by adding extra materials into the Appendix.

---

### Author Response · Authors · 2024-11-23
**Meta reply for all reviewers**

### **Meta Reply**

We sincerely thank you for your valuable comments and suggestions. We understand that your primary concern lies in how our framework compares to existing Bayesian methods for addressing uncertainties in reinforcement learning. In this paper, we addresses a fundamental limitation of Bayesian RL methods when applied to the dynamic RL processes.

**1. Misalignment Between Bayesian Philosophy and Online RL Training:**
> In online RL, an infinite stream of transition data is available. Consequently, the uncertainties of network parameters arise primarily from the stochastic nature of the environment, not from a fixed dataset. Bayesian methods, however, rely on posterior distributions conditioned on a static dataset, leading to a theoretical mismatch. This misalignment raises questions about the appropriateness of Bayesian RL methods for online RL training.

**2. Dynamic Nature of RL Processes Is Often Overlooked:**
> RL training involves two interdependent steps: policy control (actor network update) and policy evaluation (critic network update). The actor network’s parameters, which generate new data, change dynamically at every time step. Existing Bayesian RL methods often disregard this evolving nature, treating historical transition tuples in the replay memory as identically distributed. This assumption weakens their theoretical rigor and their applicability to dynamically changing systems.

**3. Challenges with Uncertainty Quantification in Existing Bayesian Methods:**
> While Bayesian methods aim to quantify uncertainties in Q-functions, the convergence of the Q-network’s distribution remains unclear. Moreover, their ability to provide reliable uncertainty quantification is often unproven through numerical experiments. For instance, in the indoor escape environment [1], popular methods like Bootstrapped DQN (BootDQN) and Distributional RL (QR-DQN) fail to correctly estimate the Q-network's  distribution, resulting in unfaithful coverage rates. Additionally, random prior networks (RPNs) [2] are unable to construct accurate confidence intervals for true Q-values, for which posterior consistency is not theoretically guaranteed.

> Using publicly available code, we conducted experiments to evaluate coverage rates and mean squared error (MSE). Results in Table 1 (table summarizing 100 experiments with mean (standard deviation)) demonstrate that existing Bayesian methods fail to provide reliable uncertainty quantification in this setting. Furthermore, current Bayesian actor-critic algorithms heavily rely on these Bayesian DQN-style methods, and introducing an additional actor network only makes the interval predictions worse.

**4. Challenges in Extending Bayesian Theory to Deep Neural Networks:**
> While Bayesian methods can theoretically provide accurate uncertainty quantification, their applicability is constrained by restrictive assumptions. For instance, Gaussian Process Temporal Difference (GPTD) and Kalman Temporal Difference (KTD) methods require linearity to uphold Gaussian assumptions, limiting their use in more complex scenarios. In the context of deep neural networks, common approaches like bootstrapping with ensemble networks increase computational complexity and memory usage significantly, making them less practical for real-world applications. Additionally, techniques such as random prior functions (RPFs) are often employed in methods like Bayesian Bellman operators to introduce stochasticity. However, these approaches lack theoretical guarantees, further questioning their reliability in Bayesian reinforcement learning.

Our framework provides the following key contributions, overcoming the limitations of Bayesian RL while maintaining theoretical rigor and practical efficiency:

**1. Unified Actor-Critic Framework with Latent Trajectories:**
> We integrate the two RL steps—policy control and policy evaluation—into a unified dynamic system. By conceptualizing transition trajectories as latent variables, we use a variant of Fisher’s identity to implicitly integrate out these trajectories during policy gradient evaluation (see Eq. (4) in our paper). This approach fully accounts for the dynamic nature of RL and provides theoretical guarantees for the convergence of both actor and critic networks. Additionally, we establish convergence for the entire RL training process, offering a rigorous foundation for dynamic uncertainty estimation.

**2. Efficient and Flexible Framework Without Architectural Changes:**
>Unlike Bayesian RL methods, which often rely on ensemble networks or Gaussian process assumptions, our latent trajectory framework is lightweight and adaptable. It requires no changes to the network architecture, making it compatible with any critic network structure. Our approach seamlessly integrates with scalable SGMCMC sampling, providing both computational efficiency and ease of implementation.

---

> ### Author Response · Authors · 2024-11-23
> **Meta reply (Continue)**
>
> **3. Rigorously define and verify the uncertainty of critic network estimation**
> > In this paper, we rigorously define the uncertainty in the critic network’s estimation in a distributional sense. Specifically, we argue that the value function estimates produced by the critic network should form a non-degenerate distribution centered around the true value function of the optimal policy. Latent Trajectory framework theoretically guarantees the critic network to converge weakly to a stationary distribution $\pi(\psi|\theta^*)$ corresponding to the optimal policy $\pi_{\theta*}$.
>
> > Moreover, our experiments demonstrate that our algorithm consistently achieves the nominal 95% coverage rate for the 95% confidence intervals of the value function. This empirical result ensuring the reliability of the uncertainty estimates produced by our algorithm.
>
> ### **Appendix**
> As shown in Table 1,  our approach achieves accurate coverage rates for the value function, demonstrating both its correctness and effectiveness; in contrast, none of the existing methods yield accurate coverage rates.
>
> **Table 1**
> | Algorithm           | MSE(V) / MSE(Q)   | coverage rate  | CI-Width       |
> |---------------------|-------------------|----------------|----------------|
> | A2C                 | 0.53527 (0.03974) | 0.489 (0.0061) | 0.413 (0.0023) |
> | LT-A2C: $\mathcal{N=10000} $            | 0.00038 (0.00001) | **0.947 (0.0004)** | 0.457 (0.0009) |
> | LT-A2C: $\mathcal{N=20000} $             | 0.00039 (0.00001) | **0.947 (0.0004)** | 0.452 (0.0010) |
> | LT-A2C: $\mathcal{N=40000} $             | 0.00033 (0.00001) | **0.947 (0.0004)** | 0.449 (0.0009) |
> | PPO                 | 0.56112 (0.04272) | 0.487 (0.0066) | 0.416 (0.0024) |
> | LT-PPO: $\mathcal{N=10000} $             | 0.00041 (0.00001) | **0.947 (0.0004)** | 0.458 (0.0009) |
> | LT-PPO: $\mathcal{N=20000} $             | 0.00038 (0.00001) | **0.947 (0.0005)** | 0.452 (0.0009) |
> | LT-PPO: $\mathcal{N=40000} $             | 0.00032 (0.00001) | **0.947 (0.0004)** | 0.449 (0.0008) |
> | DQN                 | 0.09760 (0.01291) | 0.411 (0.0198) | 0.238 (0.0060) |
> | Boot-DQN            | 0.09979 (0.01609) | 0.388 (0.0186) | 0.188 (0.0032) |
> | QR-DQN              | 0.00459 (0.00028) | 0.821 (0.0089) | 0.278 (0.0033) |
> | RPN: prior scale=0.1 | 0.03339 (0.00290) | 0.802 (0.0147) | 0.679 (0.0243) |
> | RPN: prior scale=1.0 | 0.03724 (0.00412) | 0.816 (0.0157) | 0.693 (0.0243) |
> | RPN: prior scale=5.0 | 0.03658 (0.00297) | 0.793 (0.0190) | 0.782 (0.0341) |
>
>
> **Reference**
>
> [1] Shih, F., & Liang, F. (2024). *Fast Value Tracking for Deep Reinforcement Learning*. In *The Twelfth International Conference on Learning Representations*.
>
> [2] Osband, I., Aslanides, J., & Cassirer, A. (2018). *Randomized prior functions for deep reinforcement learning*. In *Proceedings of the 32nd International Conference on Neural Information Processing Systems* (pp. 8626–8638). Curran Associates Inc., Red Hook, NY, USA.

---

### Author Response · Authors · 2024-11-30
**Reminder for rebuttal review**

Dear reviewers,

We hope this message finds you well. This is a gentle reminder to review our rebuttal and revised manuscript, which we have submitted in response to your valuable comments. In the revised version, we have included a new “Related Work” section to better address the limitations of existing Bayesian methods and to clearly highlight the contributions of our work.

If you have any further questions or require additional clarifications, please feel free to reach out. We greatly appreciate your feedback and look forward to your response.

Best Regards

---

### Meta-Review · Area_Chair_q3qL · 2024-12-10

**Metareview:**

The paper reframes actor critic methods as finding the zero of an integral equation by regarding transition tuple and value function parameters as latent variables. Some theoretical and empirical results are provided. First, I am concerned about the motivation of this work. Actor critic is not simply to find the stationary point of the actor networks. The dominating actor critic methods like TRPO, PPO are designed based on a monotonic performance improvement theorem. Even for the classical policy gradient method, the goal is still to maximize the expected total reward. Ending up with a stationary point is merely a consequence (due the complexity of nonlinear optimization), not the goal. Second, it is not clear what kind of uncertainty the authors refer to -- the authors never formally define uncertainty. In RL, we can have two entirely different uncertainties, the epistemic uncertainty and the aleatoric uncertainty. More discussion about this is needed to better place the paper in the literature. Third, the empirical improvements of LTA2C is really marginal, even just compared with a pretty weak baseline. Moreover, the selection of tasks is not well justified. It is claimed that the performance improvement is from better exploration but there is no evidence supporting this, and the used tasks are not exploration desperate.

**Additional Comments On Reviewer Discussion:**

The reviewers raised some questions and the authors subsequently provide some response. But all reviewers are still unanimously against this paper. I also read the paper myself and agree with the reviewers.

---

### Decision · Program_Chairs · 2025-01-22

Reject